# HOXA9 inhibits HIF-1α-mediated glycolysis through interacting with CRIP2 to repress cutaneous squamous cell carcinoma development

Liang Zhou [1], Yinghui Wang[1], Meijuan Zhou[1], Ying Zhang[1], Pengfei Wang[1], Xiaoxing Li[2], Jing Yang[2], Hongmei Wang [2] & Zhenhua Ding [1]

Glycolytic reprogramming is a typical feature of many cancers; however, key regulators of glucose metabolism reengineering are poorly understood, especially in cutaneous squamous cell carcinoma (cSCC). Here, Homeobox A9 (HOXA9), a direct target of onco-miR-365, is identified to be significantly downregulated in cSCC tumors and cell lines. HOXA9 acts as a tumor suppressor and inhibits glycolysis in cSCC in vitro and in vivo by negatively regulating HIF-1α and its downstream glycolytic regulators, HK2, GLUT1 and PDK1. Mechanistic studies show that HOXA9-CRIP2 interaction at glycolytic gene promoters impeds HIF-1α binding, repressing gene expression in trans. Our results reveal a miR-365-HOXA9-HIF-1α regulatory axis that contributes to the enhanced glycolysis in cSCC development and may represent an intervention target for cSCC therapy.

---

[1] Department of Radiation Medicine, Guangdong Provincial Key Laboratory of Tropical Disease Research, School of Public Health, Southern Medical University, 510515 Guangzhou, China. [2] State Key Laboratory of Oncology in South China, Collaborative Innovation Center for Cancer Medicine, Sun Yat-sen University Cancer Center, 510060 Guangzhou, China. These authors contributed equally: Liang Zhou, Yinghui Wang, Meijuan Zhou. Correspondence and requests for materials should be addressed to Z.D. (email: dingzh@smu.edu.cn)

Cutaneous squamous cell carcinoma (cSCC) is the second most common cancer with an annual incidence of over one million worldwide[1–3]. Chronic sun exposure can damage the DNA of normal keratinocytes in the epidermis, primarily via ultraviolet (UV) radiation, and lead to the development of skin cancer including cSCC[2,4]. However, the underlying molecular mechanism(s) responsible for this transition remain to be fully elucidated.

With the increase of volume, tumors constantly suffer from hypoxia owing to their poorly formed vasculature. Tumor cells therefore are required to meet their oxygen demand by adjusting their metabolic form, such as glycolytic reprogramming from oxidative phosphorylation (OXPHOS) to glycolysis[5]. As a key pro-survival mechanism, such glycolytic reprogramming endows cancer cells with at least two advantages: (1) more rapid and substantial ATP generation than oxidative phosphorylation; (2) the supply of rich substrates for anabolic metabolism of nucleic acids, lipids, and amino acids[6,7]. Notably, the efficacy of glycolytic reprogramming is so beneficial and successful that rapidly-proliferating cancer cells tend to strongly enhance the glycolysis while restricting oxidative phosphorylation, regardless of oxygen levels. This phenomenon of aerobic glycolysis in cancer cells is termed the Warburg effect[6].

The hypoxia inducible factor (HIF)-1 pathway is dominantly involved in cancer-related biological processes including hypoxic response, angiogenesis, cell cycle and glycolysis[8,9]. Oxygen-responsive HIF-1α subunit and constitutively-expressed HIF-1β subunit constitute the heterodimeric HIF-1 transcription factor which plays critical roles in cellular response to hypoxia. Oxygen-sensitive prolyl hydroxylase (PHD) family utilizing oxygen as a co-substrate and iron ($Fe^{2+}$) as a co-factor have four members, each of which possesses a unique role in regulating HIF-α levels, with PHD2 acting as the primary isoform controlling HIF-1α levels in normoxia[10]. Hydroxylated HIF-1α can be ubiquitinated by von Hippel–Lindau (pVHL) E3 ubiquitin ligase and targeted for proteasomal degradation[9]. Hypoxia-mediated oxygen deprivation inactivates PHDs and thus stabilizes HIF-1α. Although HIF-1α is known for its hypoxia-responsive feature, it is regulated by many other factors under normoxia condition such as loss of tumor suppressors, reactive oxygen species (ROS) or oncogene activation[9]. In renal cell carcinoma, loss of pVHL expression owing to genetic lesions of VHL gene (mutations or deletions) results in constitutive HIF-1 stabilization[11]. The role of reactive oxygen species (ROS) in HIF-1α stabilization is more conclusive under normoxic microenvironment[12]. In tumors, oxygen byproducts such as free radicals stabilize HIF-1α[13]. PI3K/AKT pathway activated by ROS increases HIF-1α expression and prevents its degradation through increased heat shock protein expression and nitric oxide synthase activation via phosphorylation[14,15]. It is suggested that iron depletion by ROS oxidization of the iron ($Fe^{2+}$) co-factor inhibits the activity of PHD enzymes and thus contributes to the stabilize HIF-1α[16]. Further, the growth factor TGF-β1 acts as an agonist to stabilize HIF-1α through SMAD pathway-mediated selective inhibition of PHD2 expression[17].

The accumulation of HIF-1α significantly reduces the efficiency of OXPHOS and promotes glycolysis to enhance cellular survivability under both hypoxic and normoxic conditions[9,18]. As a master regulator of glycolysis, HIF-1α orchestrates glucose transporters and rate-limiting enzymes in glucose metabolism including hexokinase 2 (HK2), glucose transporter 1 (GLUT1), and pyruvate dehydrogenase kinase 1 (PDK1)[5,9,19]. HIF-1α dimerizes with HIF-1β to bind to the hypoxia response element (HRE)[20] in transcription-regulatory regions (promoters or enhancers) of downstream glycolytic genes, e.g., HK2, GLUT1, and PDK1, to promote their expression and drive the transition of glycolytic reprogramming[21]. The excessive expression of GLUT1 observed in various types of tumors is hypothesized as being necessary to meet the huge energy requirements for cancer growth, thus suggesting GLUT1 as an indicator of carcinogenesis[22]. HK2, the first rate limiting enzyme for glycolysis, is highly expressed in various cancers, which enhances aerobic glycolysis[23]. PDK1 controls the switch of glucose metabolism from aerobic oxidation to glycolysis by inactivating the pyruvate dehydrogenase E1α subunit, converting pyruvate to acetyl-CoA to inhibit pyruvate metabolism via the tricarboxylic acid cycle[24]. Such regulatory role of PDK1 is advantageous to tumor growth because it avoids the accumulation of ROS by reducing mitochondrial oxygen consumption[25,26].

Aside from the lung, the epidermis is the only epithelium in direct contact with aerial oxygen[27]. The physiological partial pressure of oxygen is relatively low in the epidermis compared to that in other tissues, which leads to a mildly hypoxic microenvironment and potentiates the adaptive accumulation of HIF-1α in the epidermis, especially in the basal layer[28]. Further, the primary environmental risk factor for cSCC tumorigenesis, solar UV radiation, also influences the expression of HIF-1α with a biphasic effect, i.e., inhibition of HIF-1α expression in the early stage followed by promotion of HIF-1α expression in the later stage[29,30]. Consistent with this, HIF-1α expression is lower in normal skin compared with cSCC tumors, wherein it is positively correlated with the ascending grades of cSCC[31,32]. In addition, HIF-1α also plays roles in the proliferation of cSCC stem-like cells[33]. However, the mechanism of HIF-1α being regulated and its relationship with HIF-1α-mediated glycolytic reprogramming in cSCC remain under explored.

In this study, Homeobox A9 (HOXA9) was identified as the direct target of oncogenic miR-365 and significantly downregulated in cSCC tumors and cells[34]. To examine the relationship between HOXA9 and HIF-1α regulation in cSCC as suggested by subsequent bioinformatics analyses, functional and mechanistic studies were conducted to establish the tumor-suppressive and glycolysis-inhibitive roles of HOXA9 in vitro and in vivo. Our study highlights a novel miR-365-HOXA9-HIF-1α regulatory axis underlying the enhanced glycolytic metabolism and tumor development in cSCC, which provides novel intervention targets for cSCC therapy.

## Results

**HOXA9 is downregulated in cSCC with elevated miR-365.** Previous identification of aberrant onco-miR-365 expression in cSCC tumors and cells[34] suggests the importance of identifying the direct downstream targets of miR-365 for understanding its roles in oncogenesis. Therefore, we screened the downstream target genes of miR-365 by web-based algorithms. Among the downregulated genes, HOXA9 was consistently predicted to be a direct target of miR-365 by three well-cited algorithms, TargetScan[35], miRanda[36], and miRDB[37]. Moreover, as a known transcription factor in development and cancer fields, the roles of HOXA9 in cutaneous carcinogenesis remained unknown.

To assess the potential involvement of HOXA9 in cSCC development and progression, we examined the expression levels of HOXA9 in the cSCC cell lines, A431, and HSC-1, in comparison with the primary keratinocytes and control cell line, HaCaT keratinocytes. The results of western blot analysis showed that the expression of HOXA9 was downregulated in all cSCC cell lines compared with the primary keratinocytes and the HaCaT keratinocytes (Fig. 1a). Additionally, miR-365 expression was inversely correlated with HOXA9 expression in these cell lines (Fig. 1a). To check if miR-365 could be induced by serum, A431 cells and primary keratinocytes were cultured in serum-free

medium with adding or not adding serum. No significant difference of the miR-365 expression levels between serum-free group and serum-containing group could be detected by qRT-PCR in both of the two types of cells (Supplementary Fig. 1). To extend this analysis to patient tumors, we first collected and verified that miR-365 was overexpressed in randomly-selected cSCC tissue samples from patients (Fig. 1b), which is consistent

with the previous study[34]. Then, both the mRNA and protein expression levels of HOXA9 were examined in these tumor samples. As predicted, HOXA9 expression levels were significantly downregulated, which is inversely correlated with that of miR-365 in the same patient tumors (One-way ANOVA, $P < 0.001$) (Fig. 1b). HOXA9 expression was further examined in paraffin-embedded sections of 84 cSCC and 8 normal specimens

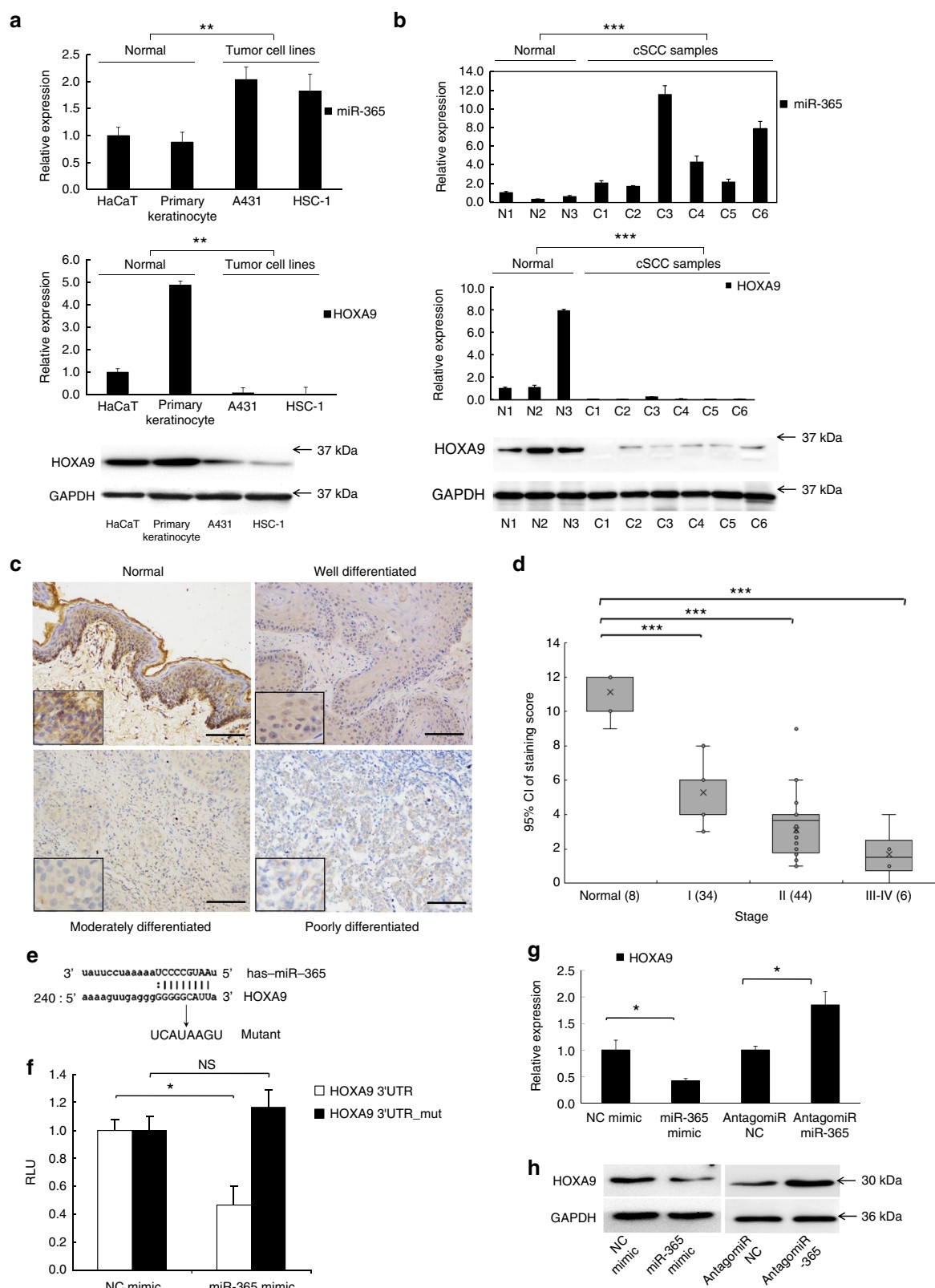

by immunohistochemistry (IHC). The results revealed low level expression of HOXA9 (weak, or no expression) in all cSCC tumors examined, whereas almost all the normal specimens showed strong HOXA9 signal (Fig. 1c and Supplementary Fig. 2). Furthermore, scoring of HOXA9 staining revealed that it was inversely correlated with ascending cSCC grades. Specifically, an evident decreasing trend was observed across normal tissue and the early stages of cSCC (grade I and II) (One-way ANOVA, $P <$ 0.05), which also extended to late stage cSCC (III & IV) (One-way ANOVA, $P < 0.05$) (Fig. 1d). RNA-seq results from another group also support that HOXA9 is significantly downregulated in cSCC tumors[38] (One-way ANOVA, $P < 0.05$) (Supplementary Fig. 3). Collectively, these results suggested that HOXA9 is downregulated in cSCC cells and primary cSCC tumors, and is inversely correlated with the expression of oncogenic miR-365.

**MiR-365 directly targets HOXA9 by binding to 3′ UTR region.** To investigate whether the downregulation of *HOXA9* is owing to direct targeting by miR-365, we searched for miR-365 binding sites within the *HOXA9* 3′UTR, revealing that miR-365 was predicted to hybridize to an evolutionarily conserved site among vertebrate species (Fig. 1e). Perfect matches exist between the seed regions of miR-365 and the 3′UTR of *HOXA9*, suggesting that miR-365 can directly repress *HOXA9* expression. This was verified by cloning the fragments of the *HOXA9* 3′UTR regions encompassing the target sites downstream of the firefly luciferase gene. Compared with NC mimic, cotransfection of miR-365 mimic together with the wild-type reporter caused significant repression of the luciferase reporter activity (Fig. 1f). Such targeting effect was specific to miR-365 binding because the reporter activity was not affected when transfections were repeated with a mutant miR-365 binding site in the *HOXA9* 3′UTR (Fig. 1f).

To address whether *HOXA9* expression was functionally related with the expression of miR-365 in cSCC cells and tumors, we predicted that miR-365 binding to the *HOXA9* 3′UTR would lead to the repression of *HOXA9*. Accordingly, ectopic expression of miR-365 by transfection of a miR-365 mimic led to decreased expression of *HOXA9*, whereas knocking down of miR-365 by antagomiR-365 could up-regulate the expression of HOXA9 at both the mRNA and protein levels (Fig. 1g, h). These results indicated that *HOXA9* represents a direct downstream target of miR-365 in cSCC cells and that knockdown of miR-365 could alleviate the repression and up-regulate the expression of HOXA9.

**HOXA9 plays anti-carcinogenic roles in cSCC.** The downregulation of *HOXA9* by miR-365 overexpression implied that HOXA9 may have an anti-carcinogenic role in cSCC tumorigenesis. To test this notion, depletion of HOXA9 was achieved by transfecting siRNA oligos against HOXA9 into A431 cells, a cell line derived from an epidermoid (squamous cell) carcinoma

(Merlino et al., 1984). Significant losses of HOXA9 at both mRNA and protein levels were achieved by siHOXA9-treated cells with respect to control (siNC) transfection (Fig. 2a and Supplementary Fig. 4). Cell proliferation was then assessed by CCK-8 assay, which showed a drastic increase of proliferative capacity after siHOXA9-treatment with respect to siNC treatment (Fig. 2b). This was supported by the results of the colony formation assay: as significantly greater numbers of colonies formed in siHOXA9-treated group than those in siNC-treated group (One-way ANOVA, $P < 0.05$) (Fig. 2c). The mobility of A431 cells was significantly enhanced upon HOXA9 knockdown as indicated by a transwell migration assay, demonstrating significantly more cells penetrated the pores of the membrane than siNC-treated group (One-way ANOVA, $P < 0.05$) (Fig. 2d). Matrigel invasiveness measurement showed that knockdown of HOXA9 also significantly promoted the invasive ability of cSCC cells (One-way ANOVA, $P < 0.05$) (Fig. 2e). Furthermore, Annexin V/ PI double staining revealed HOXA9 knockdown induced an increase in the living cell population and an accompanying decrease in the apoptotic population (Fig. 2f). Conversely, overexpression of HOXA9 in A431 cells was found to repress cSCC cell proliferation, migration, and invasiveness and promote apoptosis compared with cells transfected with empty vector (Fig. 2g–l). The pro-apoptotic function of HOXA9 was validated by treating cells with zVAD, a caspase inhibitor, which alleviated apoptosis in HOXA9-overexpressing cells (Fig. 2l).

Importantly, we did the same gain-of and loss-of-function validations in another cSCC cell line, HSC-1, and obtained similar results (Supplementary Fig. 5–6). Also, rescue experiments by restoring HOXA9 expression after HOXA9 depletion re-inhibited the enhanced proliferation, colony formation, migration, invasion, and re-promoted the apoptosis (Supplementary Fig. 7c-g). Together, the above data demonstrated that HOXA9 inhibits tumor growth, migration, and invasiveness in cSCC cells, thus warranting further elucidation of its underlying mechanism.

**RNA-Seq reveals HOXA9-regulated signaling pathways in cSCC.** To understand the HOXA9-mediated molecular events in cSCC cells, we conducted a genome-wide analysis to globally characterize HOXA9-regulated transcriptome changes. Total RNAs of A431 cells treated with siNC or siHOXA9 oligos were subjected to transcriptomic sequencing (RNA-Seq). The expression of HOXA9 mRNA and protein was validated prior to RNA-Seq (Fig. 3a). The majority of mapped reads was classified as protein coding (>70%) whereas other types of transcripts, including antisense, misc-RNA, and lincRNA were also identified (Supplementary Fig. 8). Based on the significance criterion (Q-value), a total of 3545 transcripts were found to be significantly regulated in siHOXA9-treated cells with respect to siNC-treated cells (Fig. 3b, Supplementary Data 1). Subsequent Gene Ontology analysis with the list of significantly regulated genes revealed that

**Fig. 1** HOXA9 is downregulated in cSCC cell lines and primary tumors and acts as a direct target of miR-365. The expression levels of HOXA9 and miR-365 were detected by qPCR and western blot in **a** HaCaT keratinocytes, primary keratinocytes and cSCC cell lines (A431 and HSC-1) or **b** normal skin tissues and cSCC tumors. **c** IHC detection of HOXA9 on paraffin sections of cSCC tumors and normal skin specimens. Representative images with various levels of staining (Strong from normal tissues, negative or weak from tumor tissues) are shown. Scale bar: 100 μm. (200×) **d** Association of HOXA9 IHC-staining scores with tumor grades (I, II, and III–IV). The numbers of cases are shown below. Data are plotted as the means of 95% confidence interval ± s.d. **e** Schematic illustration of base pairing between miR-365 and its putative binding sequence in the 3′ UTR of *HOXA9*. The mutated sequences are underlined. **f** Wild-type (WT) or mutant reporter constructs were transfected into A431 cells with NC or miR-365 mimics. A dual luciferase assay was performed 48 h post transfection and normalized to *Renilla* luciferase activities. Data represent the average of three independent experiments ± s.d. **g** *HOXA9* mRNA expression was measured in NC mimic, miR-365 mimic, antagomiR NC, or antagomiR-365 transfected A431 cells by qRT-PCR and normalized with *GAPDH*. Expression folds are shown with respect to NC mimic or antagomiR NC cells where normalized copy numbers were set to 1. **h** HOXA9 protein expression was measured in the above transfected A431 cells by western blot using GAPDH as a loading control. Each experiment was performed at least in triplicate and results are presented as mean ± s.d except in (**d**). One-Way ANOVA and Dunnett's multiple comparison test were used to analyze the data. (*$P <$ 0.05, **$P < 0.01$, ***$P < 0.001$)

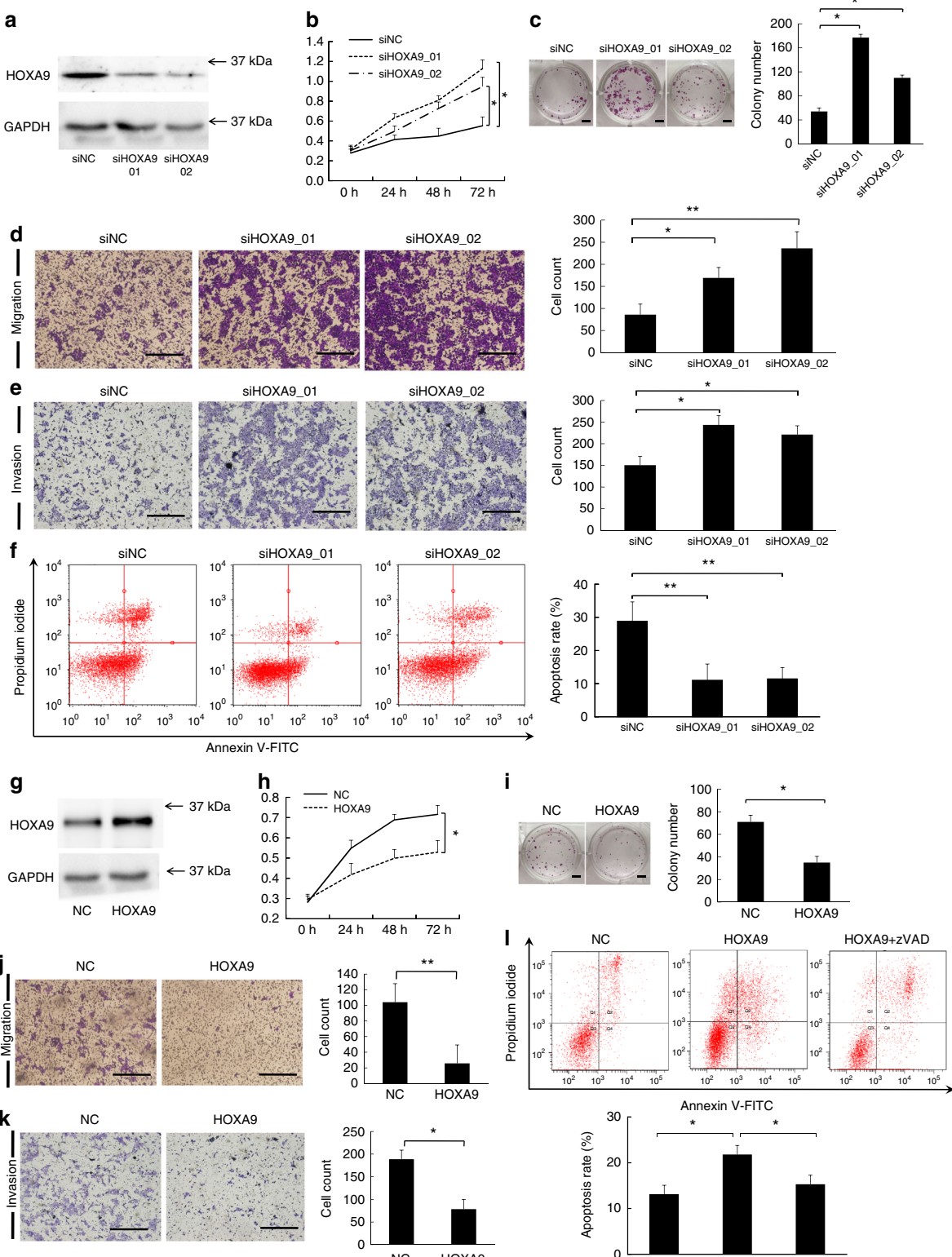

**Fig. 2** HOXA9 plays anti-carcinogenic roles in cSCC. **a** HOXA9 protein expression was detected after depletion of HOXA9 by siRNAs in A431 cells. Measurements of cell proliferation by CCK-8 assay (**b**), colony formation assay (**c**), transwell migration assay (**d**), Matrigel invasiveness measurement (**e**), and apoptosis assay by Annexin V/PI double staining (**f**) were performed in A431 cells treated with siRNAs targeting HOXA9. **g** HOXA9 protein expression was detected by western blot after overexpression of HOXA9 in A431 cells. Measurements of cell proliferation by CCK-8 assay (**h**), colony formation assay (**i**), transwell migration assay (**j**), Matrigel invasiveness measurement (**k**), and apoptosis assay by Annexin V/PI double staining (**l**) were performed in A431 cells overexpressing HOXA9. **c**, **j** Scale bar: 1 cm. **l** zVAD treatment was performed after HOXA9 overexpression to check the variation of apoptosis. Scale bar: 100 μm. Each experiment was performed in at least triplicate and results are presented as mean ± s.d. One-way ANOVA and Dunnett's multiple comparison test were used to analyze the data (*$P < 0.05$, **$P < 0.01$, ***$P < 0.001$)

the top-ranked lists of enriched Gene Ontology categories includes "Cellular metabolic process", "Cellular macromolecule metabolic process", "Macromolecule metabolic process", "Primary metabolic process", "Organic substance metabolic process", "Metabolic process", and "Glycolysis", ($P < 0.05$, Fig. 3c, Supplementary Data 2). Strikingly, Kyoto Encyclopedia of Genes and Genomes (KEGG) pathway analysis showed that important molecular pathways including "HIF-1 signaling pathway" were

significantly enhanced (Fig. 3c, d and Supplementary Data 3). Among the 32 genes affected in the HIF-1 pathway, 8 genes including *HIF-1α*, *GLUT1 (SLC2A1)*, *HK2*, *PDK1*, *PFKL*, *PGK1*, *PFKFB3*, and *ENO2* were glycolytic metabolism-related and their expression levels were significantly upregulated in response to HOXA9 knockdown as revealed by our RNA-seq and qRT-PCR (One-way ANOVA, $P < 0.05$) (Fig. 3e and Supplementary Data 1). RNA-seq results from another study strongly supported

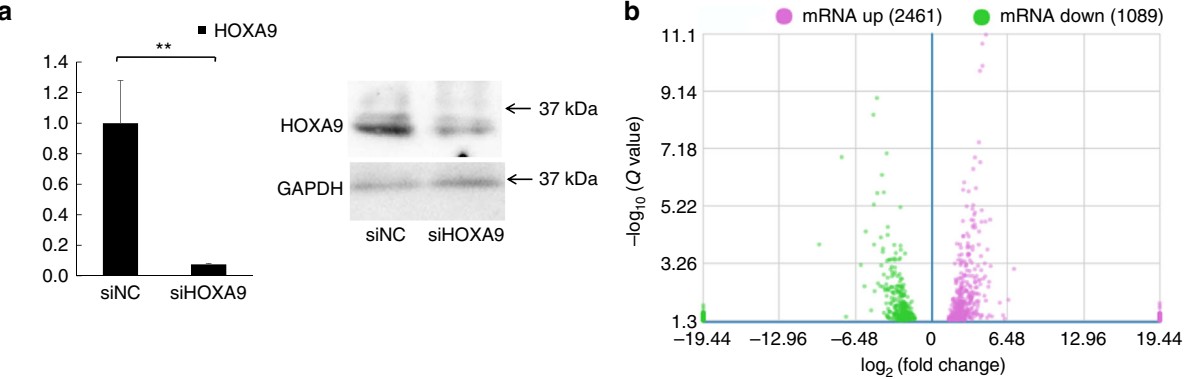

| Category | GO ID | Term | *P*-value |
|----------|-------|------|-----------|
| GOTERM_BP_ALL | GO:0044237 | Cellular metabolic process | 3.26E−23 |
| GOTERM_BP_ALL | GO:0044260 | Cellular macromolecule metabolic process | 3.77E−21 |
| GOTERM_BP_ALL | GO:0044238 | Primary metabolic process | 1.08E−17 |
| GOTERM_BP_ALL | GO:0043170 | Macromolecule metabolic process | 1.42E−16 |
| GOTERM_BP_ALL | GO:0008152 | Metabolic process | 3.42E−16 |
| GOTERM_BP_ALL | GO:0071704 | Organic substance metabolic process | 4.94E−16 |
| GOTERM_BP_ALL | GO:0061718 | Glucose catabolic process to pyruvate | 1.19E−03 |
| GOTERM_BP_ALL | GO:0061621 | Canonical glycolysis | 1.19E−03 |
| GOTERM_BP_ALL | GO:0061620 | Glycolytic process through glucose-6-phosphate | 1.67E−03 |
| GOTERM_BP_ALL | GO:0006007 | Glucose catabolic process | 5.31E−03 |
| GOTERM_MF_ALL | GO:0000988 | Transcription factor activity, protein binding | 2.28E−05 |
| GOTERM_MF_ALL | GO:0032091 | Negative regulation of protein binding | 2.34E−02 |
| GOTERM_CC_ALL | GO:0044446 | Intracellular organelle part | 4.97E−43 |
| GOTERM_CC_ALL | GO:0005634 | Nucleus | 4.13E−33 |
| KEGG_PATHWAY | hsa05200 | Pathways in cancer | 2.58E−05 |
| KEGG_PATHWAY | hsa04066 | HIF-1 signaling pathway | 4.40E−03 |
| KEGG_PATHWAY | hsa00010 | Glycolysis / Gluconeogenesis | 4.60E−03 |
| KEGG_PATHWAY | hsa05230 | Central carbon metabolism in cancer | 6.56E−03 |

Gene ontology and KEGG pathway analysis of regulated genes by HOXA9 knockdown (partial list)

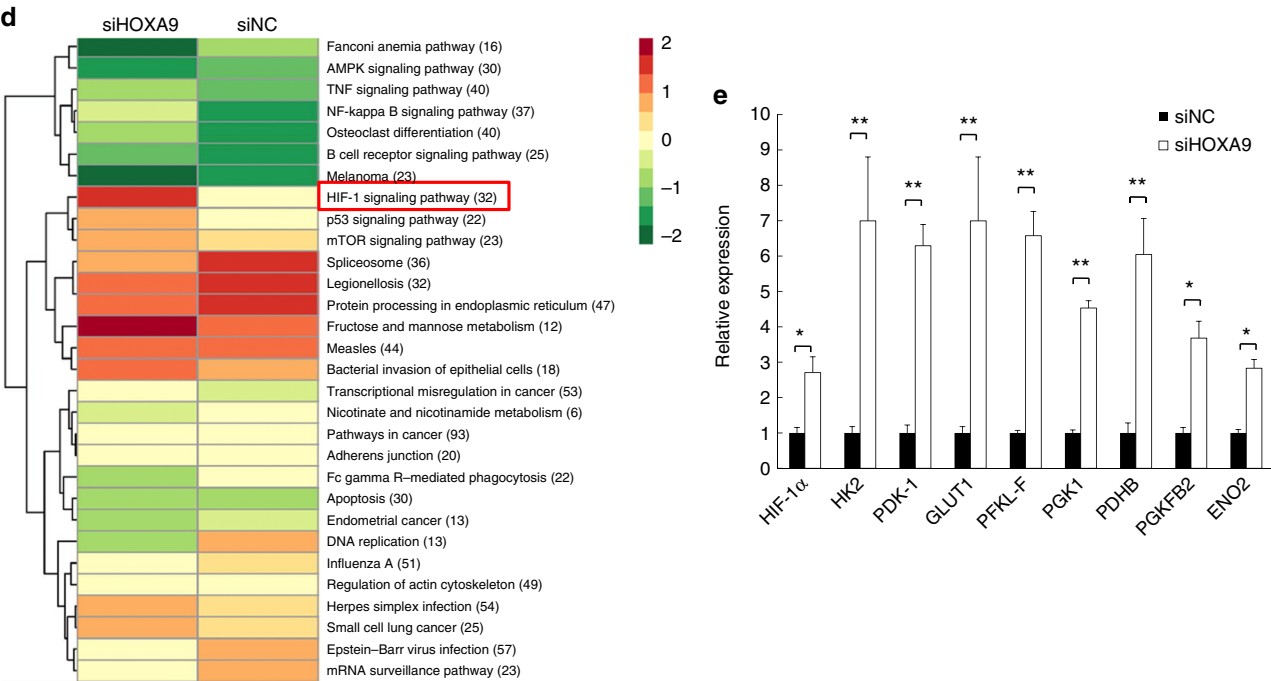

our findings that compared with Normal Human Epidermal Keratinocytes (NHEKs), HOXA9 is significantly downregulated in cSCC tumors while *HIF-1α* and its downstream glycolytic genes are mainly upregulated[38] (One-way ANOVA, $P < 0.05$) (Supplementary Fig. 3). Collectively, the above results strongly argue that HOXA9 plays a critical role in glycolytic metabolism in cSCC.

**HOXA9 represses glycolysis and promotes OXPHOS**. To access the metabolic differences (potential Warburg effect) between keratinocytes and cSCC cells (A431 and HSC-1), we conducted bioenergetics profiling of these cells using the Seahorse XF-Bioanalyzer platform. As predicted, HaCaT keratinocytes engage an OXPHOS metabolic strategy whereas cSCC cells tend to adopt glycolysis (Fig. 4a). Oxygen consumption rate (OCR) and extracellular acidification rate (ECAR) evaluation indicated that the cSCC cells consumed less oxygen (lower OCR) and generated more extracellular lactate (higher ECAR) compared with HaCaT keratinocytes (Fig. 4b). To investigate whether HOXA9 regulates the glycolytic metabolism of cSCC cells, the energy phenotype in response to the variations of HOXA9 level was analyzed. Depletion of HOXA9 enhanced the glycolytic metabolism and repressed the OXPHOS (Fig. 4c, d), whereas overexpression of HOXA9 was found to inhibit glycolysis and promote OXPHOS (Fig. 4e, f). At the same time, OCR and ECAR assays also revealed that HOXA9 performed a similar glycolysis-inhibitory role in HSC-1 cells as it did in A431 cells (Supplementary Fig. 9a–d). Rescue experiments by restoring HOXA9 expression after HOXA9 depletion re-inhibited the upregulated glycolysis owing to HOXA9 knockdown and re-established the OXPHOS levels (Supplementary Fig. 10a, b). Thus, we concluded that loss of HOXA9 promotes glycolytic metabolism in cSCC cells.

**HOXA9 inhibits the epigenetic activities of HIF-1α**. To further validate the glycolysis-regulatory function of HOXA9 supported by bioinformatics analysis of transcriptomic sequencing, HIF-1α and its downstream glycolytic genes (*HK2*, *GLUT1*, and *PDK1*) were selected from the total of 8 glycolysis-related genes owing to their critical roles in metabolic reprogramming of tumors. A431 and HSC-1 cells were treated with siNC or two different siHOXA9 oligos. Quantitative reverse transcription-polymerase chain reaction (qRT-PCR) and western blot detection was performed, which confirmed the up-regulation of HIF-1α and its downstream glycolytic genes including GLUT1, HK2, and PDK1 (Fig. 5a, Supplementary Fig. 5a,b). Conversely, overexpression of HOXA9 repressed HIF-1α and its downstream glycolytic genes (Fig. 5b, Supplementary Fig. 6a,b). Also, rescue experiments by restoring HOXA9 expression after HOXA9 depletion re-inhibited the upregulated expression of HIF-1α and its downstream glycolytic genes caused by HOXA9 knockdown (Supplementary Fig. 7a,b).

HOXA9 is known to transcriptionally regulate HIF-1α through an identified binding site[39] (Fig. 5c). To explore how HOXA9 regulates the other selected metabolic genes, the predicted promoter regions (+2000 bp to −2000 bp relative to the

transcription start site [TSS]) of *GLUT1*, *HK2*, and *PDK1* were analyzed according to a minimum consensus binding site (ATAA)[39] via the UCSC genome browser (http://genome.ucsc.edu/) and rVista (https://rvista.dcode.org/)[40] (Fig. 5c). Based on this prediction, electrophoretic mobility shift assays (EMSA) were used to detect direct associations between purified HOXA9 protein and the predicted motifs, which were also confirmed by supershift assay with a HOXA9 antibody (Fig. 5d). Notably, mutation of the core sequence of the respective binding motifs diminished the binding activities, indicating the specificity of the binding motifs for HOXA9. Chromatin immunoprecipitation (ChIP)-qPCR detection confirmed that depletion of HOXA9 by RNA interference led to marked decreases in the binding enrichment of HOXA9 at the binding sites of *HIF-1α, HK2, GLUT1*, and *PDK1* (Fig. 5e).

As previous studies have identified that HIF-1α itself could also regulate HK2, GLUT1, and PDK1[8,25], herein we predicted hypoxia response elements (HREs) and HIF-1α binding sites around the above HOXA9 sites using rVista (Fig. 5c). Strikingly, overexpression of HOXA9 in A431 cells drastically decreased the binding enrichment of HIF-1α at these HREs (Fig. 5f). Thus, the binding of HOXA9 and HIF-1α at the above loci of glycolytic genes are mutually exclusive. The up-regulation or overexpression of HOXA9 may therefore replace the HIF-1α to repress the expression of glycolytic genes including *HK2*, *GLUT1*, and *PDK1*, thereby inhibiting glycolysis to drive the conversion from glycolysis to adopt OXPHOS as a metabolic strategy.

**CRIP2 plays a tumor-suppressive function**. HOXA9 can act as either an oncogene in hematologic malignancies[41,42] or a tumor suppressor in carcinomas[43]. As HOXA9 was identified as a tumor suppressor in cSCC from the above results, it is critical to understand the underlying mechanism for the anti-carcinogenic role of HOXA9. Hence, an immunoprecipitation experiment was performed with a HOXA9 antibody and a non-specific IgG control. The specific binding partners of HOXA9 from the pulled-down protein complexes were detected by comparing the differential binding profiles identified with high performance liquid chromatography-mass spectrometry (HPLC-MS) (Fig. 6a, Supplementary Fig. 11 and Supplementary Table 1). Among the potential binding partners of HOXA9, CRIP2 is an anti-carcinogenic transcription factor[44,45] that fits the supposed role of an HOXA9 interaction partner. CRIP2 was also verified to exist in the protein mix immunoprecipitated by HOXA9 using western blot (Fig. 6b). Notably, their direct interaction was additionally supported by the finding that HOXA9 overexpression led to enhanced pull-down of CRIP2 protein, which indicates a specific and dose-dependent direct interaction (Fig. 6c).

**The glycolysis-inhibitory role of HOXA9 is CRIP2-dependent**. To study the role of CRIP2 in cSCC, a loss-of-function study by RNA interference was performed. CRIP2 knockdown led to the significant upregulation of HIF-1α, HK2, GLUT1, and PDK1 protein expression (Fig. 7a) and enhanced the cell proliferation, migration, and invasiveness of A431 cells (Fig. 7b–d). Notably,

**Fig. 3** Genome-wide analysis of HOXA9-regulated transcriptomic changes by RNA-Seq in cSCC cells. **a** The expression of HOXA9 mRNA and protein was verified in A431 cells treated with siRNA targeting HOXA9 by qRT-PCR and western blot. Each experiment was performed in triplicate and data are presented as mean ± s.d. One-Way ANOVA and Dunnett's multiple comparison test were used to analyze the data (*$P < 0.05$, **$P < 0.01$, ***$P < 0.001$). **b** Total RNAs were isolated from A431 cells treated with siNC or siHOXA9 oligos and subjected to high throughput mRNA sequencing. Differentially expressed genes between siNC-treated and siHOXA9-treated A431 cells were determined by RNA-Seq and shown by volcano plot. **c** Over-represented Gene Ontology categories by Gene Ontology analysis of differently-expressed genes. BP biological process, MF molecular function, CC cellular component, KEGG Kyoto Encyclopedia of Genes and Genomes. The HIF-1 signaling pathway is highlighted. **d** Differentially expressed genes were clustered and shown in a heat map. Color bars at the right represent gene clusters established through k-means clustering. **e** Validation of identified glycolytic genes in the HIF-1 pathway including *HIF1A, HK2, GLUT1, PDK1, PFKL, PGK1, PFK2, PDHB,* and *ENO2*

loss of CRIP2 led to enhanced glycolysis and repression of OXPHOS as shown by OCR and ECAR analysis (Fig. 7e, f and Supplementary Fig. 12a, b). To functionally verify that CRIP2 is necessary for the glycolysis-inhibitory role of HOXA9, the complementary experiment by overexpressing HOXA9 after CRIP2

depletion was performed. The results showed that HOXA9 no longer suppressed the expression of HIF-1α and its downstream glycolytic genes when CRIP2 was depleted (Fig. 7g). This finding was strengthened by the findings from ECAR and OCR analyses, which showed that loss of CRIP2 eliminated the function of

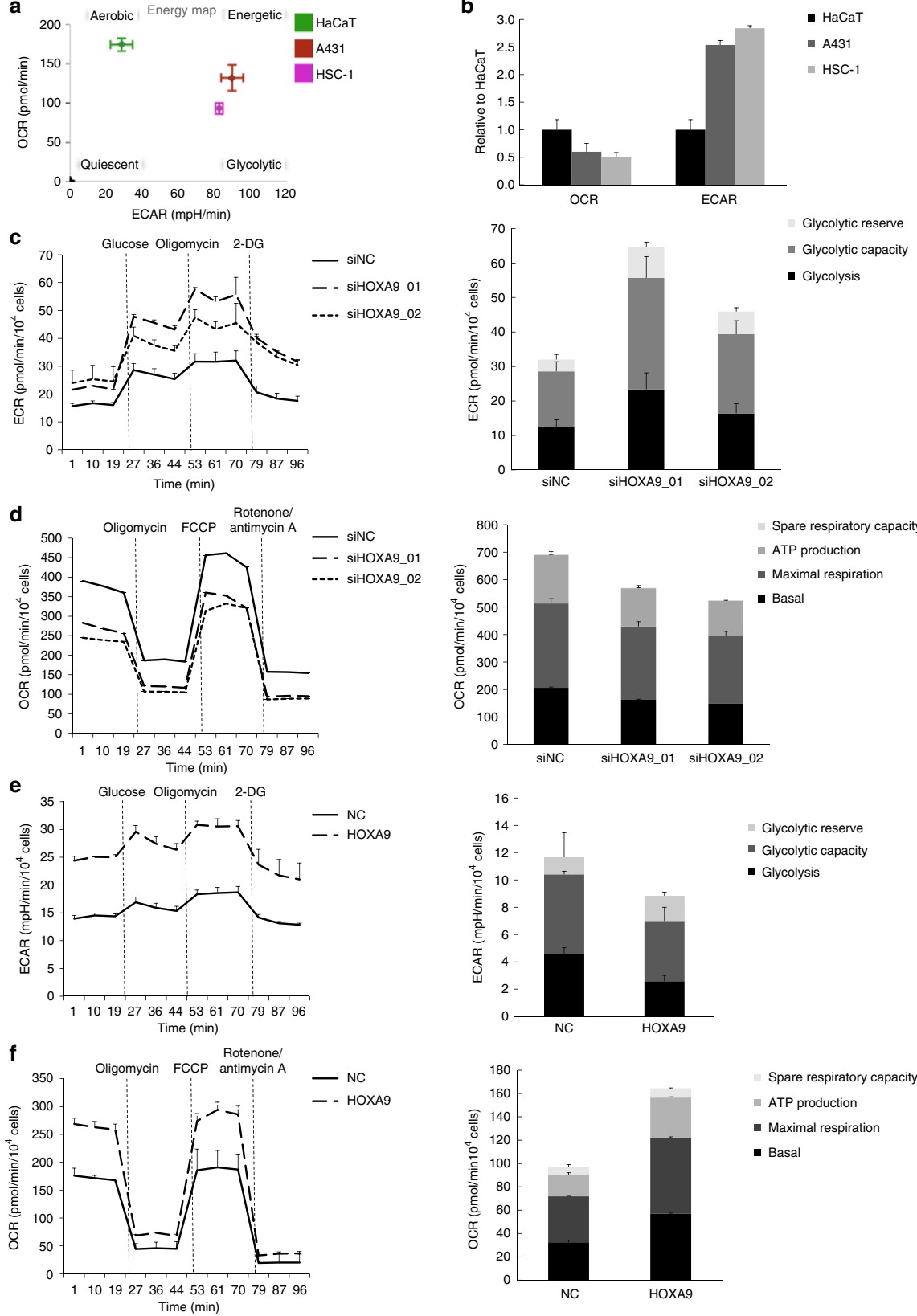

HOXA9 in metabolic regulation (Fig. 7h, i and Supplementary Fig. 12c, d). Together, these findings support the notion that the glycolysis-suppressive roles of HOXA9 are CRIP2-dependent.

**Loss of HOXA9 enhances glycolysis in vivo**. To evaluate the anti-carcinogenic effect of HOXA9 in vivo, a xenograft tumor model was established in immunocompromised mice. No difference in tumor sizes was initially apparent between the two groups. After the 9th day, HOXA9 depletion markedly enhanced tumor growth when compared with the siNC control group. Tumor size and mass were evidently larger at the end of the evaluation period (Fig. 8a, b). Loss of HOXA9 expression in the siHOXA9 group was confirmed by qPCR, western blot, and IHC staining (Fig. 8c–e). Furthermore, HOXA9 depletion led to the significant upregulation of HIF-1α and its downstream glycolytic genes, including GLUT1, HK2, and PDK1 (Fig. 8c–e). To explore the role of HOXA9 in metabolism regulation, we evaluated glucose metabolism in siNC-treated and siHOXA9-treated xenograft tumors by tail vein injection of $^{18}$F-Fluordeoxyglucose ($^{18}$F-FDG). Micro-positron emission tomography (microPET)/computed tomography (CT) imaging clearly showed enhanced glucose uptake and retention in HOXA9-depleted tumors (Fig. 8f).

Furthermore, a HOXA9 expressing vector or empty vector control was injected into xenograft tumors. HOXA9 overexpression markedly delayed tumor growth immediately following injection when compared with the vector control (Supplementary Fig. 13a). Tumor size and mass were evidently smaller at the end of evaluation (Supplementary Fig. 13b). HOXA9 overexpression significantly down-regulated the expression of HIF-1α and its downstream glycolytic genes as shown by qRT-PCR, western blot and IHC staining (One-way ANOVA, $P < 0.05$) (Supplementary Fig. 13c–e). microPET/CT imaging of $^{18}$F-FDG indicated that overexpression of HOXA9 strongly inhibited glucose uptake and retention (Supplementary Fig. 13f). Collectively, the in vivo experiments indicated that HOXA9 repressed glycolysis by negatively regulating HIF-1α and its downstream glycolytic genes including GLUT1, HK2, and PDK1.

## Discussion

Emerging evidence supports that cancer is primarily a metabolic disease involving dysfunction in energy production mediated via respiration and fermentation[46]. The genomic instability observed in tumor cells as well as other recognized hallmarks of cancer are considered to represent downstream epiphenomena of the initial disturbance of cellular energy metabolism[5]. Thus, it is important to elucidate the metabolic relationship between known oncogenes (including onco-miRs) and cancers, especially in cSCC, a cancer type with few explorations in metabolism.

The HOX gene family encodes a series of transcription factors that play key regulatory roles in embryonic development; studies have also shown them to be associated with cancers[47]. HOXA9 regulates embryonic development and functions in different stages of various tumors, either as an oncogene or as a tumor suppressor. Notably, rearrangement of mixed lineage leukemia (MLL), associated with a recurring chromosomal abnormality that occurs in acute myeloid leukemia (AML) and acute

lymphoblastic leukemia is affiliated with poor prognosis and abnormal expression of the HOX gene family, especially HOXA9. Studies have confirmed that HOXA9 expression is closely related to the abnormal proliferation of myeloid cells, which contributes to the survival and development of leukemia[41,42]. HOXA9 is also critical to maintain stem cell self-renewal and avoid aging of normal hematopoietic stem cells and leukemia stem cells[48]. In addition to the effects of aberrant expression, HOXA9 mutations or the formation of fusion genes such as NUP98-HOXA9, can cause AML[49]. HOXA9 can also inhibit MEIS1-mediated apoptosis[50] and the anti-apoptotic BCL-2 may act as the key factor for the HOXA9-dependent survival of leukemia cells[51]. Thus, HOXA9 may act as an oncogene in hematologic malignancies.

However, in solid tumors, the role of HOXA9 is not unique. In breast cancer, HOXA9 and its downstream target BRCA1 are significantly downregulated, which was thought to be critical for the development of breast cancer in patients lacking estrogen receptor/progesterone receptor expression[43]. Downregulation of HOXA9 was suggested as the indicator for the occurrence of non-small cell lung cancer whereas overexpression of HOXA9 can significantly repress the invasiveness of A549 and HCC95 cells[52]. In ovarian cancer, HOXA9 transcriptionally activates TGF-β2 to stimulate the proliferation of fibroblasts for the formation of a suitable microenvironment and promotes cancer progression[53]. Therefore, no universal model exists to predict the exact role of HOXA9 in a cancer type. HOXA9 may act as a tumor suppressor or an oncogene, depending on the context of specific cancers.

In this study, HOXA9 was identified to play an anti-carcinogenic role, inhibiting cell proliferation, migration, and invasiveness in cSCC, while promoting cell apoptosis. Among the routine in vivo and in vitro evaluations of HOXA9 functions, when using xenograft tumor model, there is an inflection in tumor growth after around 9 days in the absence of HOXA9. Checking our previous studies[54,55], we found that manipulating the gene expression in xenograft tumors by transfection generally takes about 9 days to observe the visible volume change. In contrast, in vitro cellular experiments require only 2–3 days to observe significant changes. Our results indicate the differential sensitivities between in vivo and in vitro models in response to the variations of specific gene expression.

As HOXA9 plays a pro-carcinogenic role in blood cancer, identification of the factor that mediates the "switching" of HOXA9 as an oncogene or a tumor suppressor would be of particular value to understand and potentially treat disparate oncogenesis.

Over the past decade, with the emergence of a large amount of evidence supporting the critical roles of glycolytic metabolism in cancers, researchers have started to view cancer as a metabolic disorder[46]. Notably, the glycolysis dominant in most tumors is aerobic glycolysis rather than the common glycolysis enhanced only under hypoxic conditions in normal cells[5]. The regulatory mechanism of glycolytic flux in normal cells is subverted in cancer cells to inhibit rate-limiting steps and reengineer the glucose metabolism by inducing many glycolytic enzymes and transporters, such as HK2, GLUT1, and PDK1[5,19]. Thus, the regulatory mechanisms enhancing the fermentation of glucose,

**Fig. 4** HOXA9 represses glycolysis and promotes oxidative phosphorylation. **a** Metabolic phenotypes of keratinocyte and cSCC cell lines were assessed by the ratio of OCR/ECAR. **b** OCR and ECAR levels of the above cell lines were plotted as fold change relative to HaCaT. **c, d** Left: ECAR and OCR analysis of A431 cells, in which HOXA9 was depleted using two siRNAs, followed by treatment with the indicated compounds; Right: glycolytic variations (glycolysis, glycolytic capacity, and glycolytic reserve) or OXPHOS variations (basal, maximal respiration, ATP production, and spare respiratory capacity) were summarized from raw data. **e, f** Left: ECAR and OCR analysis of A431 cells, in which HOXA9 was overexpressed, followed by treatment with the indicated compounds; right panels: glycolytic variations (glycolysis, glycolytic capacity, and glycolytic reserve) or OXPHOS variations (basal, maximal respiration, ATP production, and spare respiratory capacity) were summarized from raw data. Each experiment was performed in triplicate and data are presented as mean ± s.d. One-Way ANOVA and Dunnett's multiple comparison test were used to analyze the data (*$P < 0.05$, **$P < 0.01$, ***$P < 0.001$)

e.g., promoting the expression of the glycolytic enzymes and transporters, are critical for our understanding of the development of specific cancers including cSCC.

Although HIF-1α acts as a key factor regulating hypoxia response and glucose metabolism in cancer biology, the relationship between HOXA9 and HIF-1α–mediated hypoxia response and metabolic regulation has rarely been explored, especially in cSCC. HOXA9 has been reported to coordinate with both MEIS1 and PBX1 to promote HIF-1 expression in hematopoietic stem cells (HSCs)[39]. It is no surprising that HOXA9

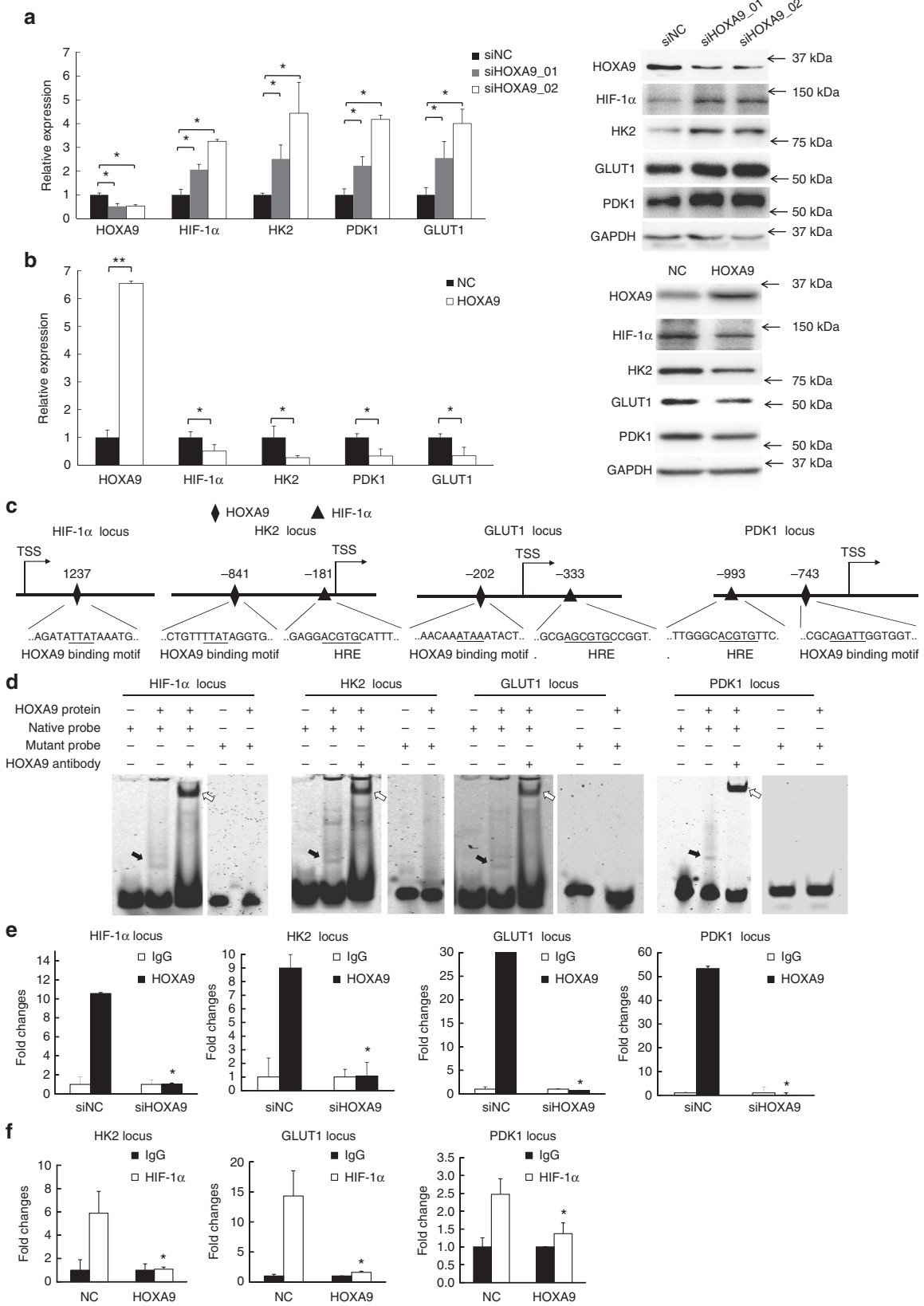

always acts as an oncogene in hematologic malignancies and positively regulate pro-carcinogenic HIF-1 expression. However, in cSCC, our findings clearly demonstrate that HOXA9 plays tumor-suppressive roles and negatively regulates the expression of HIF-1α and its downstream glycolytic genes. Notably, HOXA9 restricts glucose uptake as shown by our in vivo [18]F-FDG metabolic experiments, which indicate the glycolysis-inhibitory nature of HOXA9 by repressing HIF-1α and its downstream glycolytic genes in cSCC.

However, HOXA9 plays the exact opposite role in cSCC compared with that in HSCs. This finding also raised the question of whether a factor exists that functions in "regulating" or "switching" the role of HOXA9 between pro-glycolysis and anti-glycolysis.

CRIP2, a cysteine-rich protein that contains two LIM domains, was originally identified to be critical for cardiovascular development by bridging serum response factor and GATA proteins and stimulating smooth muscle target gene expression[56]. Recently, CRIP2 was shown to act as a tumor suppressor. In nasopharyngeal carcinoma, loss of CRIP2 promotes tumorigenesis and angiogenesis by interacting with NF-κB/p65 and compromises NF-κB-mediated proangiogenic cytokine expression including IL6, IL8, and VEGF[44]. Another group identified CRIP2 to be significantly downregulated by aberrant expression of miR-449a in breast cancer. Notably, stable overexpression of CRIP2 in MDA-MB-231 breast cancer cells inhibited tumor growth and angiogenesis through repression of the NF-κB/p65-VEGF axis[57]. Additionally, CRIP2 has been shown to induce apoptosis in esophageal squamous cell carcinoma by activating caspase 3 and 9 proteins. In summary, the studies support a tumor-suppressor function for CRIP2[45].

Herein, CRIP2 was identified to be a direct interacting partner of HOXA9 and acted as a tumor suppressor in cSCC, consistent with the previous-reported anti-carcinogenic role of CRIP2[44,45,57]. Notably, HOXA9 complementation experiments after CRIP2 knockdown indicated that the glycolysis-inhibitory function of HOXA9 is dependent on CRIP2. This finding highlights the critical role of CRIP2 in HOXA9-mediated glycolytic reprogramming and may also potentiates the underlying mechanism for the opposite roles of HOXA9 in different cancers.

Evidence of the mechanisms driving the Warburg effect in cancers has provided good examples to explain involvement of miRNA regulation in energy metabolism. The initial step in energy metabolism is the entry of glucose into cells through glucose transporters (GLUTs). Downregulation of miR-1291 in renal cancer, miR-195-5p in bladder cancer, and miR-150 in pancreatic cancer leads to enhanced expression of their direct targets, GLUT1, GLUT3, and GLUT4 respectively, which thus results in glucose uptake and contributes to tumor progression[58–60]. In the following step, the first rate-limiting enzyme of glycolysis, HK2, is directly targeted by miR-143 in lung, colon, and prostate cancers[61,62]. Let-7 is commonly downregulated in many cancers,

which leads to the upregulation of its direct target, PDK1, and facilitates aerobic glycolysis[63]. Notably, miRNAs are involved in regulating almost the entire process of glucose metabolism[64].

However, studies regarding the roles of miRNAs in cSCC are scarce, especially in terms of glucose metabolism. Previous studies showed that miR-365 was highly expressed in cSCC tumors and cell lines, and this miRNA was identified as an UV-induced onco-miR that played critical roles in the malignant transformation of keratinocytes. Treatment with antagomiR-365 could inhibit cutaneous tumor formation in vivo[34,65]. Explorations regarding the downstream target genes, e.g., IL6, cyclin D1, CDC25A, thyroid transcription factor 1, and nuclear factor I B have primarily unveiled the regulatory mechanism of miR-365 in different cancers[54,66–70]. Here, the role of miR-365 was extended to encompass a novel downstream target, HOXA9. Bioinformatics analysis of the HOXA9-depleted transcriptome in cSCC cells uncovered the new role of HOXA9 in regulating glycolytic metabolism, which is critical for the tumor-suppressive function of HOXA9 as confirmed by the subsequent validations. This finding extends our understanding of the function of oncogenic miR-365 to the field of tumor energy metabolism.

Collectively, our findings establish a novel miR-365-HOXA9-HIF-1α regulatory axis that functions in promoting HIF-1α-mediated glycolytic reprogramming in cSCC. As depicted in Fig. 8g, HOXA9 is downregulated in both cSCC cell lines and primary tumors owing to aberrantly-expressed miR-365. Loss of HOXA9 upregulates HIF-1α and the downstream glycolytic genes of the HIF-1 pathway, which contributes to glycolytic reprogramming and leads to cSCC progression. In normal skin cells or HOXA9-reconstituted cSCC cells, HOXA9 interacts with CRIP2 and epigenetically represses HIF-1α expression, which leads to the replacement of HIF-1α by the HOXA9-CRIP2 complex at the promoter regions and inhibits the expression of glycolytic genes, such as HK2, GLUT1, and PDK1. This epigenetic replacement of HIF-1α by HOXA9 is critical for subverting tumor cell metabolism from aerobic glycolysis to OXPHOS and the inhibition of tumor growth. Our findings highlight the tumor-suppressive and glycolysis-inhibitory roles of HOXA9 in skin tissue, support our previous finding that miR-365 is an onco-miR for the progression of cSCC, and emphasize a newly identified miR-365-HOXA9-HIF-1α axis that may provide novel intervention targets for cSCC therapy.

## Methods

**Patient samples**. This study was approved by the Institutional Review Board of Nanfang Hospital affiliated to Southern Medical University, and all patients provided written informed consent for the use of surgical samples. cSCC samples were obtained from patients diagnosed with cutaneous SCC from January 2009 to August 2016 in the departments of dermatology, pathology, and oncology at Nanfang Hospital and Zhujiang Hospital, affiliated to Southern Medical University. Briefly, a total of 84 specimens of cSCC (53 samples from men and 31 from women) and 8 specimens from normal skin were included in the analysis. Approximately 92% of the patients were more than 50 years of age. Fresh samples

**Fig. 5** HOXA9 represses the expression of HIF-1α and its downstream glycolytic genes by directly binding to the promoter regions. **a**, **b** The mRNA or protein expression of HOXA9, HIF-1α, HK2, GLUT1, and PDK1 was detected by qRT-PCR or western blot, respectively, in A431 cells after depletion of HOXA9 by two siRNAs or overexpression of HOXA9. The qRT-PCR data were normalized to *GAPDH* gene expression. In western blots, GAPDH was used as a loading control. **c** Known or predicted binding sites of HOXA9 (diamond) or HRE elements for HIF-1α (delta) in the promoter regions of *HIF1A*, *HK2*, *GLUT1*, and *PDK1* as determined by rVista (https://rvista.dcode.org/). **d** Electrophoretic mobility shift assay (EMSA) was used to detect the direct association of purified HOXA9 protein with its binding sites on the promoter regions of *HIF1A*, *HK2*, *GLUT1*, and *PDK1*. The black arrows indicate the binding complex of the HOXA9 protein with the probe containing the known or predicted binding sites whereas the white arrows indicate the supershift generated by the association of the anti-HOXA9 antibody with the above complexes. Also, the specificity of the direct associations was supported by the loss of binding activities using mutated probes. **e** The binding enrichment of HOXA9 at the above binding sites on the promoter regions of *HIF1A*, *HK2*, *GLUT1*, and *PDK1* was detected by ChIP-PCR after knockdown of HOXA9. **f** The binding enrichment of HIF-1α at the above binding sites on the promoter regions was detected after overexpression of HOXA9. Each experiment was performed in triplicate and data are presented as mean ± s.d. One-Way ANOVA and Dunnett's multiple comparison test were used to analyze the data (*$P < 0.05$, **$P < 0.01$, ***$P < 0.001$)

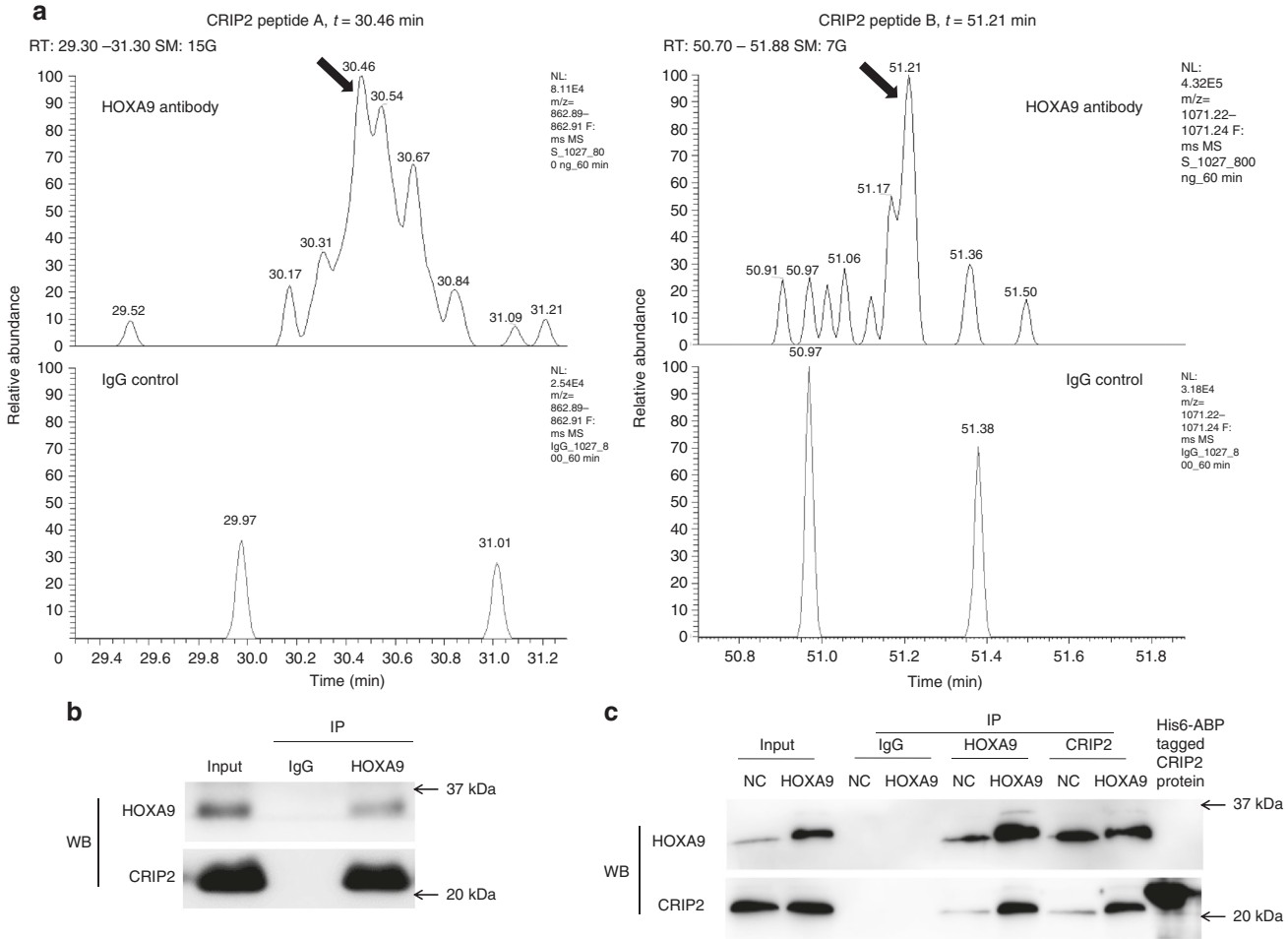

**Fig. 6** HOXA9 represses glycolytic genes by association with CRIP2. **a** The chromatographic analysis of the protein mixes immunoprecipitated by HOXA9 antibody or control IgG. Arrows indicate the identified CRIP2 peptide peaks in the HOXA9-pulldown sample, which were lacking in control IgG sample. **b** CRIP2 was verified to be the interacting partner of HOXA9 by Co-IP followed with western blot detection. IP immunoprecipitation, WB western blot. **c** Co-IP of both HOXA9 and CRIP2 using specific antibodies was performed and followed with western blot detection to confirm their direct interactions. Each experiment was performed at least in triplicate

obtained during surgery were immediately frozen in liquid nitrogen for subsequent total RNA extraction, protein extraction, and paraffin embedding. Tumors were classified according to the SCC Broders Pathological Classification (Cassarino et al. 2006): grade I (well differentiated) with 75–100% differentiated cells, grade II (moderately differentiated) with 50–75% differentiated cells, and grades III and IV (poorly differentiated) with 0–50% differentiated cells. Of the 84 cases, 34 were well differentiated (40.5%), 44 were moderately differentiated (52.4%), and 6 were poorly differentiated (7.1%).

**Animal studies**. This study was approved by the Institutional Animal Care and Use Committee (IACUC) of Nanfang Hospital affiliated with the Southern Medical University (Approval code L2016103). They are in accordance with the guidelines of the Asian Federation of Laboratory Animal Science Associations (AFLAS) and the National Regulations for the Administration of Affairs Concerning Experimental Animals (8 January 2011). Mouse transportation, housing, and breeding were conducted according to the recommendations of "The use of non-human animals in research." Male athymic nude mice (BALB/C-nu/nu, 4–5 weeks old), purchased from the animal center of Southern Medical University, were used for xenograft studies. The mice were euthanized by cervical dislocation to prevent suffering.

**Cell lines**. cSCC lines A431 (Female, CellCook Biotech Co. Ltd.) and HSC-1 (Male, HonSun Biological Co. Ltd.), the human benign epidermal keratinocyte cell line HaCaT (Male, CellCook Biotech Co. Ltd.) and Normal Human Epidermal Keratinocytes (PromoCell, Co. Ltd.) were cultured in Dulbecco's modified Eagle medium (DMEM, Life Technologies) supplemented with 10% fetal bovine serum (Invitrogen) or serum-free keratinocyte growth medium 2 and maintained at 37 °C with 5% $CO_2$ in a humidified atmosphere. The authentication information of all

the cell lines used in this study were listed in Supplementary Fig. 14. All the cell lines have been tested and shown to be no mycoplasma contamination.

**RNA isolation and quantitative real-time PCR**. Total RNAs from cells were extracted using TRIzol reagent (Life Technologies) according to the manufacturer's instructions. cDNAs were prepared using Moloney murine leukemia virus Reverse Transcriptase (Life Technologies) and Oligo(dT)20 primer. mRNA expression analysis was performed using SYBR Green Master Mix (Life Technologies) on a LightCycler 96 Detection System (Roche) using *GAPDH* for normalization. Primers used in this study are listed in Supplementary Table 2.

**DNA constructs**. The HOXA9 expressing construct was purchased from Addgene (http://www.addgene.org) with a signed material transfer agreement.

**3′-UTR luciferase reporter assay**. Validation of miRNA targets was performed by cloning partial *HOXA9* 3′-UTRs containing the sequence recognized by the miR-365 core seed region. The oligos carrying native and mutated miR-365 binding sites with SpeI and SacI restriction sites for cloning were annealed and cloned into the pMIR-reporter vector (Promega). The oligos used in this study are listed in (Supplementary Table 2).

For dual luciferase reporter assays, HEK293T cells were cotransfected with pMIR-reporter constructs, the *Renilla* luciferase reporter vector (Promega), and either 150 nM miR-365 or control mimic (Guangzhou Ribobio), using Lipofectamine 2000 reagent (Life Technologies), according to the manufacturer's instructions. Luciferase activity was measured at 48 h after transfection using the Dual-Luciferase Reporter Assay System (Promega), according to the manufacturer's protocol. Correction for differences in transfection

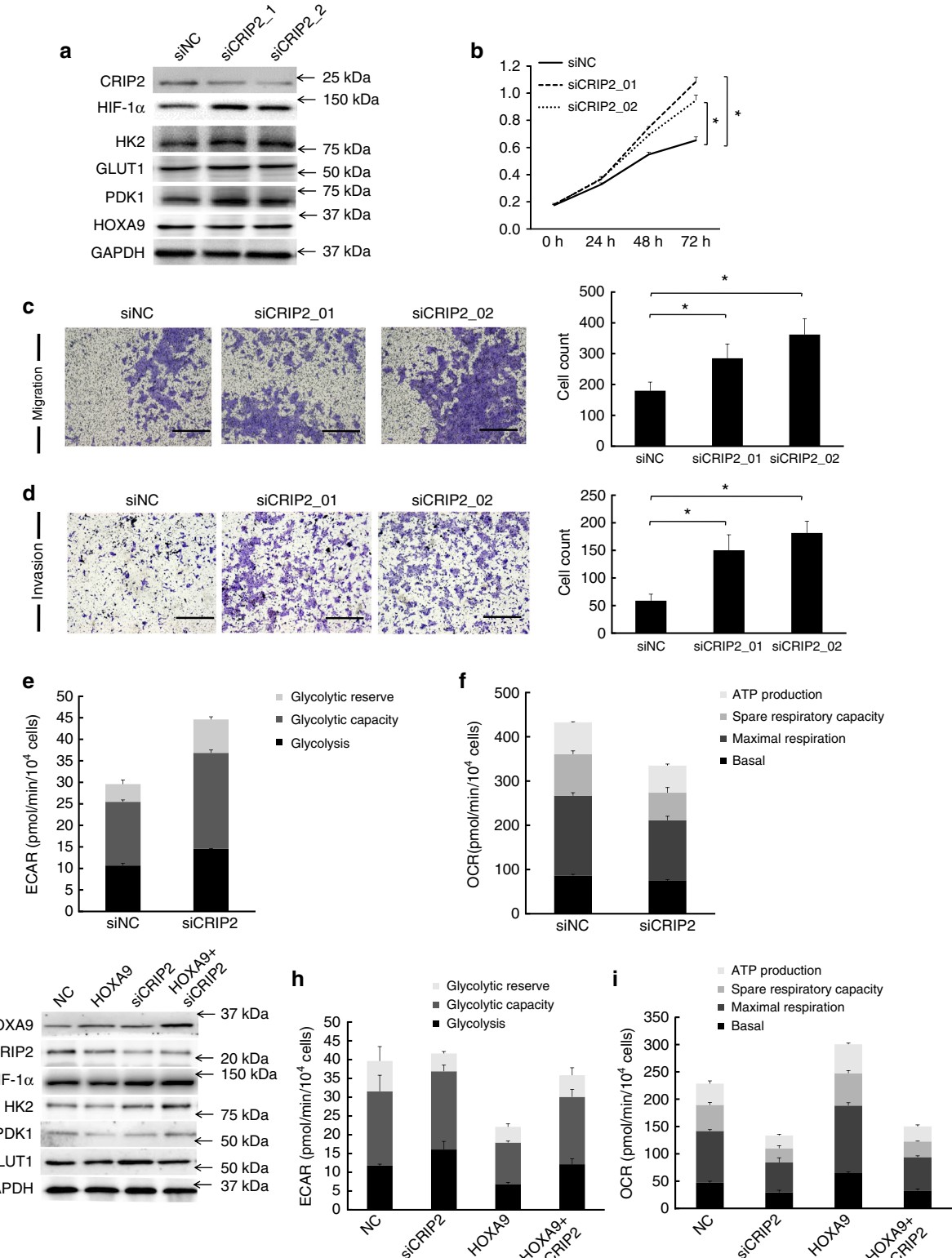

**Fig. 7** The glycolysis-inhibitory role of HOXA9 is dependent on CRIP2. **a** Western blot detection showed that CRIP2 knockdown led to significant upregulation of HIF-1α, HK2, GLUT1, and PDK1, whereas HOXA9 expression was not affected. **b–d** Depletion of CRIP2 enhanced cell proliferation, migration, and invasiveness of A431 cells by CCK-8 assay, transwell migration assay, and Matrigel invasiveness measurement. **e, f** ECAR and OCR analysis of A431 cells following depletion of CRIP2 by siRNA as summarized from raw data. **g** Western blot detection showing the variations of protein expression levels of HOXA9, CRIP2, HIF-1α, HK2, GLUT1, and PDK1 in response to CRIP2 knockdown and/or HOXA9 overexpression. **h, i** ECAR and OCR analysis of A431 cells depleted of CRIP2 by siRNA and/or overexpressing HOXA9 as summarized from raw data. Each experiment was performed in triplicate and data are presented as mean ± s.d. One-Way ANOVA and Dunnett's multiple comparison test were used to analyze the data (*$P < 0.05$, **$P < 0.01$, ***$P < 0.001$)

efficiency was performed by normalizing firefly luciferase activity to total *Renilla* luciferase.

**RNA-Seq**. Sequencing was performed at Beijing Novogene Co., Ltd. using the Illumina HiSeq 2500 instrument. RNA-Seq data was aligned to the reference genome (human assembly GRCh37/hg19) using Tophat2 (http://ccb.jhu.edu/software/tophat). HTSeq (http://www-huber.embl.de/HTSeq) was then applied on the aligned data set to determine differentially expressed genes with a "significant" status. The Gene Ontology and KEGG analyses of the differentially expressed genes were performed using DAVID (https://david.ncifcrf.gov/).

**Immunoblotting and IHC assays**. Total cell extracts were prepared and assayed by western blot as previously described (Zhou et al. 2012a, b). The following primary antibodies and dilutions were used: HOXA9 (Abcam, ab140631, 1:2000) and

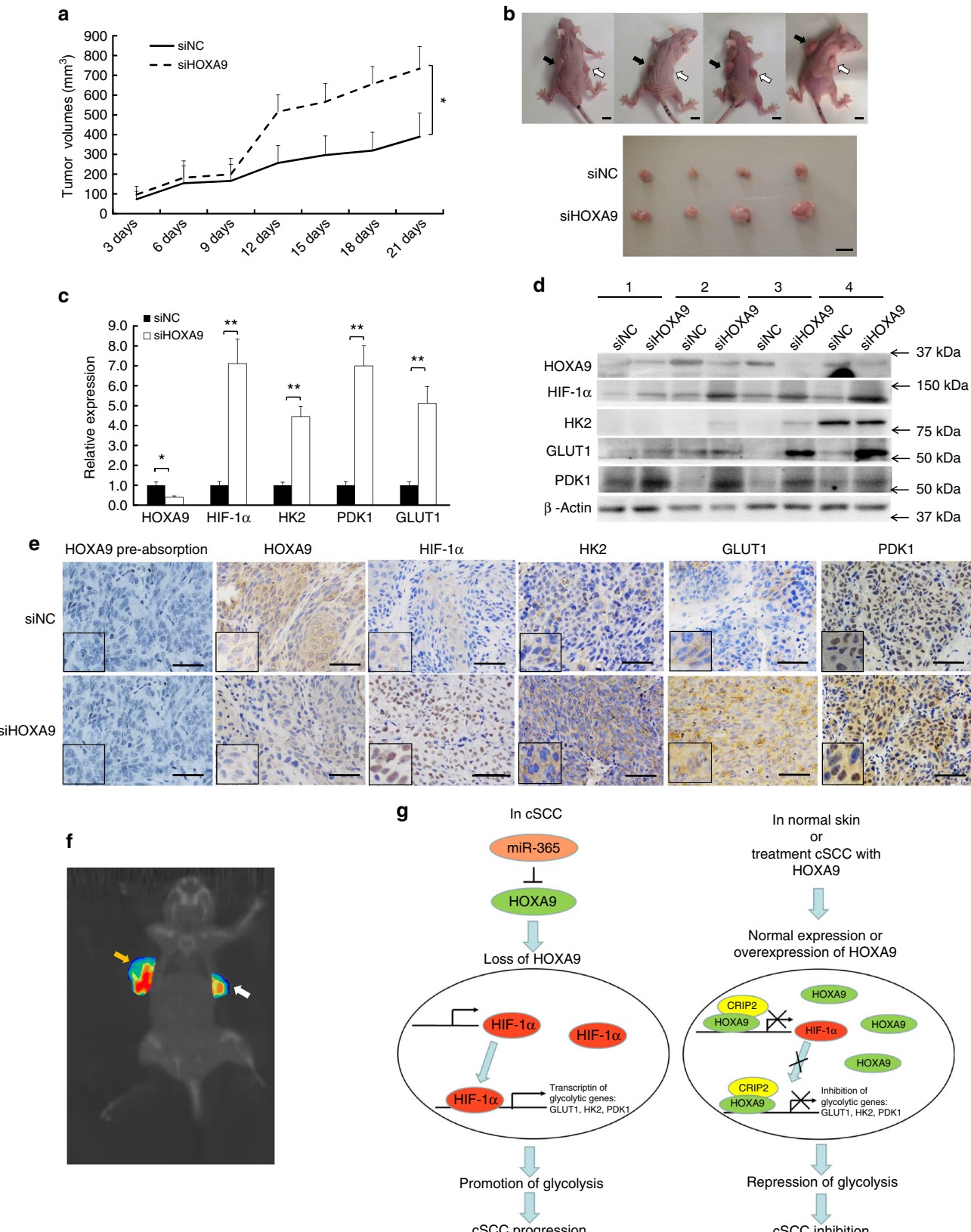

CRIP2 (Abcam, ab151496, 1:2000); HIF-1α (Santa Cruz Biotechnology, sc-10790, 1:2000), HK2 (Santa Cruz Biotechnology, sc-374091, 1:2000), GLUT1 (Cell Signaling Technology, 12939S, 1:2000), and PDK1 (Cell Signaling Technology, 3062, 1:2000); and α-TUBULIN (Santa Cruz Biotechnology, sc-5286, 1:2000) and GAPDH (Santa Cruz Biotechnology, sc-25778, 1:5000). The following secondary antibodies were also used: anti-mouse IgG-horseradish peroxidase (HRP), anti-rabbit IgG-HRP, and anti-goat IgG-HRP (Santa Cruz Biotechnology). Bound antibodies were visualized with the Luminata Forte Western HRP substrate (Millipore). Uncropped images of the blots are shown in Supplementary Fig. 15.

IHC staining of formalin-fixed paraffin-embedded cSCC sections or xenograft tumor sections was performed with the following antibodies: HOXA9 (1:100), or HIF-1α (1:100), HK2 (1:100), GLUT1 (1:100), or PDK1 (1:100) from Cell Signaling Technology. To verify the specificity of HOXA9 detection, HOXA9 antibody was preabsorbed with HOXA9 protein (Abcam, ab18030). Before IHC staining the mixture of protein and antibody was incubated in PBS with 1% BSA for 24 h at 4 °C. The ratio of antibody and protein is 1 mol:20 mol. Stained IHC sections were imaged with a ZEISS Axio Vert.A1 microscope and at least 10 representative images were collected for statistical analysis. The IHC staining was performed "blind" with respect to the different treatments.

**HwPLC-MS analysis.** A 20 μg sample of immunoprecipitated protein mix was separated by sodium dodecyl sulfate-polyacrylamide gel electrophoresis (SDS-PAGE) and stained with Coomassie brilliant blue R250 and then processed with the In-Gel Tryptic Digestion Kit (Cat. No.89871). The resulting digest was treated with Zeba Spin Desalting Columns (Part No. 89882) then subjected to analysis by Thermo Fisher Scientific orbitrap fusion LC-MS/MS in positive ion, linear, delayed-extraction mode. Calibration was carried out using a standard peptide mixture. The mass spectra were subjected to sequence database search with Proteome Discoverer v2.1 software (Thermo Scientific).

**Xenograft mouse model.** Briefly, male athymic nude mice (BALB/C-nu/nu, 4–5 weeks old) were adopted. $1.0 \times 10^7$ cells were subcutaneously implanted into the left and right flanks of nude mice. At 8 days after implantation, siNC or siHOXA9 oligos were injected into the left or right tumor, respectively; and the injection was repeated every other day. The experiments were performed "blind" with respect to the different treatments. Oligos were prepared by pre-incubating 3 nM siRNA per mouse with Lipofectamine 2000 (Life Technologies) for 15 min; injections were made using a final volume of 50 μl in serum free DMEM. The tumor diameters were measured and recorded every day to generate a tumor growth curve. After tumor growth assessment, the tumors were excised and snap-frozen for RNA and protein extraction or paraffin-embedded for IHC staining.

**ChIP-qPCR analysis.** ChIP experiments were performed as recommended by the EZ ChIP™ Chromatin Immunoprecipitation kit manual (Cat. no. 17-371, Merck Millipore) using 5 μg antibodies against HOXA9 (CST) or isotype IgG (Merck Millipore) used as a negative control. Genomic DNA pellets were purified using phenol chloroform extraction and ethanol precipitation, and then resuspended in 20 μl water. qRT-PCR was performed with 1 μl immunoprecipitated material using the SYBR Select Master Mix (Life Technologies). Relative enrichment was calculated as the amount of amplified DNA normalized to input and relative to values obtained after normal IgG immunoprecipitation, which were set as 1. Primers used are listed in Supplementary Table 2.

**Electrophoretic mobility shift assay.** EMSA experiments were carried out using forward strand 5′Cy5-labeled-dsDNA (Life Technologies). DNA probes were prepared by mixing equimolar amounts of complementary strands in 1× annealing buffer (20 mM Tris-HCl, pH 8.0, 50 mM MgCl₂, and 50 mM KCl), heated to 95 °C for 5 min and subsequently with 1 °C min⁻¹ ramping down to 4 °C in a PCR block.

Typical binding reactions contained 100 nM dsDNA with varying concentrations of HOXA9 proteins (Abcam, ab152453) in a 1× EMSA buffer [20 mM Tris-HCl pH 8.0, 0.1 mg ml⁻¹ bovine serum albumin (Bio-Rad), 50 mM ZnCl₂, 100 mM KCl, 10% (v/v) glycerol, 0.1% (v/v) Igepal CA630, and 2 mM β-mercaptoethanol] and were incubated for 1 h at 4 °C in the dark to reach binding equilibrium. Reactions were loaded into a pre-run 6% native 1× Tris-glycine (25 mM Tris pH 8.3; 192 mM glycine) polyacrylamide gel and DNA complexes were separated at 4 °C for 30 min at 200 V. The bands were detected using the ODYSSEY infrared imaging system (LI-COR Biosciences). Uncropped images of the gels are shown in Supplementary Fig. 15.

**Co-IP assay.** Pierce crosslinking magnetic IP/Co-IP kit (Thermo Scientific) was used. Prior to immunoprecipitation, binding of antibody (5 μg) to protein A/G magnetic beads was performed by incubating for 15 min at room temperature according to the manufacturer's protocols. Disuccinimidyl suberate (20 μM) was used for crosslinking the bound antibody. Cells were lysed at 4 °C in ice-cold IP Lysis/Wash buffer and cell lysates were cleared by a brief centrifugation (12,000×g, 10 min). Concentrations of proteins in the supernatant were determined by Bradford assay. Samples containing equal amount of proteins were incubated with various irrelevant IgG or specific antibodies in the presence of protein A/G magnetic beads for 1 h on a rotator. After incubation, protein A/G magnetic beads were washed extensively with IP Lysis/Wash buffer and proteins were eluted by boiling in 5× SDS sample buffer before SDS-PAGE.

**Cell proliferation assay.** A431 cells (4000 per well) cultivated on 96-well plates were transfected with siRNAs and cell proliferation was detected after 0, 24, 48, and 72 h using a cell counting kit (TransGen Biotech) at 450 nm as described in the manual. For the colony forming assay, transfected cells were incubated in 6-well plates at 1000 cells per well, which were maintained in DMEM. Medium was replaced 2 times. At day 7, cells were collected after being washed twice with PBS and fixed in 4% paraformaldehyde for 30 min. Finally, the cells were stained with 0.1% crystal violet. Visible colonies were photographed and counted.

**Apoptosis assay.** A431 cells were seeded on a 60-mm dish and transfected with siNC, siHOXA9_01, or siHOXA9_02 and cultured for 48 h. TransDetect Annexin V-FITC/PI cell apoptosis detection kit (TransGen Biotech) was applied according to instructions. Cell apoptosis was detected and quantified using a Guava easyCyte Flow Cytometry System (Merk Millipore) by the sum of both bottom and top right quadrants of the plots, e.g., including both early and late apoptotic cells. Caspase inhibitor Z-VAD-FMK (50 μM, Selleck) treatment was performed in cSCC cell lines (A431, HSC-1) for 48 h after transfection of HOXA9-expressing plasmid.

**Transwell assay.** To assess cell migration, $2.0 \times 10^5$ A431 cells transfected with siNC or siHOXA9 were seeded into the 8-μm upper chambers of 12-well plates (Merk Millipore) in serum-free DMEM. During culture at 37 °C for 48 h, the cells in the upper chambers were attracted by the culture medium in the lower chamber, through chemoattractants provided by the included 10% fetal bovine serum. The chambers were washed with PBS twice and fixed with 3.7% formaldehyde. Cells were permeabilized using 100% methanol at room temperature, stained with 0.1 crystal violet, and observed using a microscope after the cells remained in the wells being scraped off with cotton swabs.

**Matrigel invasiveness assay.** For the assessment of invasive ability, Matrigel-coated chambers (Merck Millipore) were used to culture transfected A431 cells, of which $1.0 \times 10^5$ cells were seeded into the upper chambers. Other treatments were performed as in the migration assay.

---

**Fig. 8** Loss of HOXA9 leads to enhanced glycolysis and tumor growth in vivo. siNC and siHOXA9 oligos were injected into A431 cell xenografts every 3 days. **a** Loss of HOXA9 promotes subcutaneous tumor growth in a mouse xenograft model. Tumor volumes (mm³) were plotted according to day. Tumor volume statistical data represent the average of four independent experiments ± s.d, respectively. **b** The mice were sacrificed at the end of the experiment and images taken along with the dissected tumors from five representative mice are shown. White arrows indicate the siNC-treated xenografts whereas black arrows indicate siHOXA9-treated xenografts. Scale bar, 1 cm. **c** The expression of *HOXA9*, *HIF1A*, *HK2*, *GLUT1*, and *PDK1* was measured in the dissected tumors by qRT-PCR. qRT-PCR statistical data represent the average of four independent experiments ± s.d. **d** The protein expression of HOXA9, HIF-1α, HK2, GLUT1, and PDK1 was detected in xenografts after siHOXA9 treatment by western blot. **e** Histopathology analysis (IHC staining) of HOXA9, HIF-1α, HK2, GLUT1, and PDK1 on tumor sections. HOXA9 pre-absorption tests was also performed to validate the specificity of HOXA9 antibody. Scale bar, 100 μm (200×). **f** Comparison of glucose consumption between siHOXA9-treated and siNC-treated xenograft tumors by microPET/CT imaging of the uptake and retention of ¹⁸F-FDG injected via the tail vein. A representative microPET/CT image is shown. **g** A model of the miR-365-HOXA9-HIF-1α glycolysis-regulatory axis in cSCC development. In cSCC tumors, loss of HOXA9 up-regulates HIF-1α and its downstream glycolytic genes of *HK2*, *GLUT1*, and *PDK1* in the HIF-1 pathway, which contributes to the enhancement of glycolysis and promotes cSCC progression. In normal skin or HOXA9-treated cSCC, HOXA9 interacts with CRIP2 and epigenetically represses HIF-1α expression, which leads to the replacement of HIF-1α by the HOXA9-CRIP2 complex at the promoter regions and represses the expression of glycolytic genes including *HK2*, *GLUT1*, and *PDK1*, which subsequently contributes to the inhibition of tumor progression. Each experiment was performed in triplicate and data are presented as mean ± s.d. One-Way ANOVA and Dunnett's multiple comparison test were used to analyze the data (*$P < 0.05$, **$P < 0.01$, ***$P < 0.001$)

**Statistical analysis**. Statistical tests were performed for independent-samples with an unpaired t-test or one-way ANOVA tests (SPSS version 17.0, SPSS Inc.). All statistical tests incorporated two-tailed tests and homogeneity of variance tests, and were considered to reflect significant differences if *$P < 0.05$, **$P < 0.01$, or ***$P < 0.001$. Details of statistical analyses including sample numbers ($n$) are included in the respective figure legends.

**Data availability**. RNA-seq data have been deposited in NCBI GEO datasets with the following accession codes: GSE100144, GSM2671914 and GSM2671915.

All data generated or analysed during this study are included in this published article and its Supplementary files and available from the corresponding author on request.

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

## Acknowledgements

The work described in this paper was substantially supported by grants from the National Natural Science Foundation of China (Grant No. 81573076, 81172634, 81772914, 81673105, and 81472922; http://www.nsfc.gov.cn/); a grant from Guangdong Provincial Department of Science and Technology, China (Grant No. 2016A030313738; http://www.gdstc.gov.cn/), and grants from the School of Public Health of Southern Medical University, China (Grant No. GW201612; http://web2.fimmu.com/phatm/).

## Author contributions

Z.D. and L.Z. conceived and designed the experiments. L.Z., Y.W., M.Z., Y.Z., P.W., X.L., J.Y., and H.W. performed the experiments. L.Z. and Y.W. analyzed the data. Z.D., L.Z., and Y.W. wrote the paper. Z.D., L.Z., and Y.W. reviewed and edited the manuscript.

## Additional information

**Competing interests:** The authors declare no competing interests.

