## [Peer Review File · Nature Communications]

Reviewers' comments:

Reviewer #1 (Remarks to the Author):

The manuscript by Zhou and colleagues examines the role of HOXA9 in regulating HIF1a to alter the metabolic status of cutaneous squamous cell carcinoma.

The authors show that HOXA9 inhibits glycolysis in cSCC tumors and cell lines by repressing HIF1a. HOXA9 associates with CRIP2 to inhibit HIF1a binding to target genes. The data suggest that this mir365/HOXA9/HIF1a pathway regulates glycolysis in cSCC or at least in A431 cells.

Overall, this is an interesting paper and appears to add new and significant data regarding regulation of glycolysis in cSCC. However, there are a number of significant issues that need to be addressed to improve the manuscript.

The following issues should be addressed:

Major: the authors should use an additional cSCC line than A431. A431 is very unusual in that its EGFR levels are 200-440 fold that of primary keratinocytes. This is due to genetic amplification of the locus. This upregulation is not typical of cSCC but more typical of oral/head and neck SCC. Most cSCCs do not amplify the EGFR locus like oral SCCs.

The IHC data for HOXA9 do not seem clear. Controls demonstrating specificity of staining should be performed.

1) line 69, p 4, point number 2 is a bit confusing as worded, perhaps better to say that enzymes for anabolic metabolism are upregulated, provision is a non-specific term.

2) line 98-the assertion that most oxygen for the skin is atmospheric is false. In clinical conditions where the local blood supply is blocked the skin dies and cannot survive from atmospheric oxygen. This study is based on data not relevant to in vivo metabolism of skin. The assertion contradicts the known pathophysiology of a number of vasculitic and vasculopathic skin diseases-this should be removed.

3) Fig 1A-the authors should add primary human keratinocytes as a negative control. Is the level of mir365 expression dependent on the presence of serum?

4) Fig 1C-the epidermis in the Normal panel needs to be oriented to the top of the field. There seems to be little histologic difference between moderate and low differentiation. Low is not a conventional term: cSCCs are typically well, moderately or poorly differentiated. The low differentiated panel is not clear-is this supposed to be poorly differentiated? Please provide H+E stained panels.

What is the staining pattern for HOXA9-membrane, cytoplasmic or nuclear or a combination?

5) The western blot data indicate variable HOXA9 protein levels. Why are the cSCCs negative on IHC-IHC is more sensitive than western blotting.

6) The absence of staining in 1C does not correlate with the presence of staining in 1D.

7) line 214 significance rather than significant

8) Gene ontology analysis typically gives very low p values, much lower than 2×10^{-2} . Why is the HIF1a pathway p value not that strong in fig 3D

9) Fig 4-a number of letters are missing

10) line 284-85 better to say hematologic malignancies and carcinomas rather than blood cancers or solid cancers

11) Figs 6a and b could be supplemental.

12) Fig 6C and D need a CRIP2 peptide control to determine if the association is specific.

13) The downregulation of HOXA9 by siCRIP2 in fig 6G is not that significant. How does motility increase in 6C and D with so little change in HOXA9.

14) Fig 8e-the staining for HOXA9 could be non-specific. Please clearly show that binding is specific with a HOXA9 peptide preabsorption control.

Reviewer #2 (Remarks to the Author):

The manuscript by Zhou and colleagues reports interesting findings, given the description of a novel regulatory axis involved in a well-established tumor metabolic condition, aerobic glycolysis. The authors proved with consistency the relationship between a known-to-be oncogenic microRNA in various types of cancer, miR-365; HOXA9, a cancer context-specific developmental gene; and HIF-1 α , the hypoxia-inducible factor extensively described to be up-regulated in several cancers directly affecting tumor cells' glycolytic capacity. The hypothesis was proved, in summary, by gene and protein expression analysis, cellular functional experiments (proliferation, migration, invasion and apoptosis) and metabolic functional experiments (OCR and ECR), analyzed in CSCC patient samples and cell lines, and performed in vitro and in vivo. Gene silencing and overexpression were appropriately applied to corroborate results, in silico analyses were correctly performed for functional predictions and protein-binding assays were rightly used to confirm members of the regulatory axis. Importantly, authors proved by MicroPET/CT imaging of 18F-FDG-treated tumors in vivo changes in glucose consumption rates with silencing/overexpression of HOXA9. That is a solid experiment to wrap up the explanation of the hypothesis proposed.

Some comments to be addressed

1 The rescue experiments for the target of miR-365 should be included, in order to prove that the target is really important for the phenotype proposed. This is important as one microRNA has hundreds/thousands of functionally important targets.

2 Could the authors use some databases of CSCC expression profiling to check if the members of the interactor network have correlated expressions in the way the mechanistic model predicts? This is also an independent assessment of their data obtained in the set of samples presented here.

3 Western blot result shown on Figure 1A has an extra sample (HSC-5), but no results. Were experiments performed in this cell line also?

4 Figure 2A shows protein expression levels results only, whereas line 185 of manuscript describes "...losses of HOXA9 both mRNA and protein levels...". Gene expression analysis (qPCR results) need to be included in this Figure.

Thank you for the opportunity to review this manuscript.

Reviewer #3 (Remarks to the Author):

The manuscript from Zhou and colleagues describes a novel link between the homeobox gene, HOXA9, metabolic reprogramming and cancer development. The study provides overall high quality evidence for the mechanism by which HOXA9 can suppress HIF1A activation of glycolytic gene targets through its interaction with CRIP2. This is apparently distinct from the mechanism identified previously in hematopoietic stem cells, whereby HOXA9 co-operates with other binding partners to increase HIF-mediated glycolytic gene activation (Kocabas et al. Cell Biosci 2015). It would be important for the authors to discuss their results more equally alongside this previous publication, rather than the slightly biased handling of the paper (page 17, lines 399-402). Although overall the evidence is sufficient to support this interaction and activity of HOXA9. However, some results need further refinement. Specific points are:

1. In the introduction, the authors discuss hypoxia and HIF1 stabilization (page 4, final para), particularly with reference to the aetiology of skin cancer (page 5, final para). This is likely to confuse the non-expert reader, due to the fact that this manuscript is essentially concerned with the study of HIF1 in normoxia - i.e. activated by other mechanisms. Although it is accepted that reference to hypoxia is the easiest way to introduce HIF initially, it would perhaps be more relevant for the content of the manuscript if non-hypoxic means of stabilization the alpha subunit were introduced rather than hypoxic.
2. GAPDH has previously been shown to be hypoxia-responsive in a number of cell types and is not suitable as a 'housekeeping gene' in hypoxia studies (Zhong et al. Biochem Biophys Res Commun [1999]). If the authors wish to use this single housekeeping gene, they must show unambiguously that its expression is not altered in their systems.
3. Data from Seahorse experiments is variable and not convincing. For example, in Figure 4E and G, the authors show that the mitochondria in the control cells are already highly uncoupled (compare untreated and FCCP) and the majority of oxygen consumption is not through the respiratory chain (Rotenone/Antimycin A). This is not consistent with the data in Supplementary Figure 2B, in which a more expected result is shown for the same condition (comparing siNT from each experiment). This suggests that their results are not consistent between experiments, which is concerning and brings into doubt the results they show upon intervention.
4. In Figure 2F, if apoptosis is hypothesized then this needs to be confirmed with caspase inhibitor, such as zVAD.
5. The knockdown shown in Figure 5A is not more than 50% - it is surprising that this reduction in mRNA can create such a decrease in protein, shown on the right. Is there an effect on mRNA stability as well?
6. Figure 5C - the PDK1 blot is not convincing as the band is hardly visible
7. The immunoprecipitation experiment, shown in Figure 6D appears not to have worked in the case of the CRIP2 pull-down - as there is little signal for either CRIP2 or HOXA9.
8. The inflection in tumor growth after around 9 days in the absence of HOXA9 (Figure 8A) is very interesting - could the authors discuss this further in the discussion?
9. The knockdown of HOXA9 is also unconvincing in Figure 8E - indeed, the image shown appears to have increased HOXA9. The staining in general in this figure is confusing - HK2 is cytosolic (appears to be nuclear), GLUT1 should be plasma membrane, and PDK1 is mitochondrial - although this wouldn't be visible, it shouldn't be strongly nuclear.

Minor points:

1. Figure 1H blots are not convincing - on the right, both loading control and HOXA9 increase after AntagomiR expression
2. Is the same total cell number represented in each of the scatterplots shown in Figure 2F, as if not, it is difficult to compare?

3. No Figure legend is shown in Figure 2H
4. Contrary to what the authors wrote on page 4, para 2, cancer cells in the main do not 'adopt glycolysis instead of oxidative phosphorylation'. Both are used as without OXPHOS, the appropriate level of anabolism would not be sustained, despite glycolysis. As an example, see papers by Birsoy et al. and Sullivan et al. Cell (2015).
5. Sentences on page 5 that start, 'PDK1 can inactivate...' (line 91) and 'PDK1 also controls...' line 94 appear to be saying the same thing, and could be simplified.
6. It is a little too over-generalized to say that 'Over the past decade, cancer researchers have started to view cancer as a metabolic disorder' (page 17, para 2).

Response to Reviewers' comments:

We greatly appreciate your willingness to consider our revised manuscript, and we especially thank the reviewers for their constructive comments. We have made every effort to address their concerns by carefully proofreading the manuscript to eliminate errors and ambiguities, re-writing several parts of the manuscript, and providing some relevant unpublished data. We also have our manuscript for language polishing by a professional scientist.

We provide below the point-by-point responses to the comments of reviewers, which includes explanations of how we have revised the manuscript to clarify ambiguity and address concerns. For the ease of the reviewers, we have also included quotes from the altered text when appropriate.

We hope that our changes have now made the manuscript suitable for publication. We are more than happy to provide quick response to address any further question.

Reviewer #1 (Remarks to the Author):

Q: Major: the authors should use an additional cSCC line than A431. A431 is very unusual in that its EGFR levels are 200-440 fold that of primary keratinocytes. This is due to genetic amplification of the locus. This upregulation is not typical of cSCC but more typical of oral/head and neck SCC. Most cSCCs do not amplify the EGFR locus like oral SCCs.

A: Thanks for referee's constructive comments. We have done a full functional analysis of HOXA9 using an additional cSCC cell line, HSC-1, to validate our finding and got similar results as what we have obtained using A431 cell line. Please find these results in

Supplementary Fig. 5, 6 (for the role as a tumor suppressor) and Supplementary Fig. 7 (for the glycolysis-inhibitive role), which are also emphasized in the main text.

Q: The IHC data for HOXA9 do not seem clear. Controls demonstrating specificity of staining should be performed.

A: Thanks for referee's comments. The pre-absorbed controls demonstrating the staining specificity were performed and added to Fig. 8e and Supplementary Fig. 13e.

Q: 1) line 69, p4, point number 2 is a bit confusing as worded, perhaps better to say that enzymes for anabolic metabolism are upregulated, provision is a non-specific term.

A: Thanks for referee's suggestion. Here, what we mean is that glycolysis provides rich substrates for the synthesis of nucleic acids, lipids, and amino acids. So, this sentence has been modified to be "2) providing rich substrates for anabolic metabolism of nucleic acids, lipids, and amino acids."

Q: 2) line 98-the assertion that most oxygen for the skin is atmospheric is false. In clinical conditions where the local blood supply is blocked the skin dies and cannot survive from atmospheric oxygen. This study is based on data not relevant to in vivo metabolism of skin. The assertion contradicts the known pathophysiology of a number of vasculitic and vasculopathic skin diseases-this should be removed.

A: Thanks for referee's instruction. The mentioned sentence "Oxygenation of the skin epidermis occurs mainly from atmospheric oxygen rather than the vasculature." has been removed.

Q: 3) Fig 1A-the authors should add primary human keratinocytes as a negative control. Is the level of mir365 expression dependent on the presence of serum?

A: Thanks for referee's suggestion. We added primary human keratinocytes as controls and the results has been added to Figure 1A. The HOXA9 expression in primary human keratinocytes is indeed higher than cSCC cells. We also checked the expression levels of miR-365 in both A431 cells and primary keratinocytes with or without the presence of serum. The results were placed in Supplementary Fig. 1. In light of our results, we conclude that the expression level of miR-365 is not dependent on the presence of serum.

Q: 4) Fig 1C-the epidermis in the Normal panel needs to be oriented to the top of the field. There seems to be little histologic difference between moderate and low differentiation. Low is not a conventional term: cSCCs are typically well, moderately or poorly differentiated. The low differentiated panel is not clear-is this supposed to be poorly differentiated? Please provide H+E

stained panels.

What is the staining pattern for HOXA9-membrane, cytoplasmic or nuclear or a combination?

A: Thanks for referee's instruction. First, the epidermis in the Normal panel has been oriented to the top of the field. We are sorry not providing typical images. We reviewed all of our IHC results and updated the mentioned images to provide clear histological differences among the different grades (Figure 1c). Limited by the space in Figure 1, the H & E stained images were provided in Supplementary Fig. 2. The terms used for describing the differentiation grades have been corrected to the terms suggested by the referee. To clarify the staining pattern of HOXA9, we checked our previous staining results and performed new IHC staining, all of which show the staining pattern of HOXA9 is a type of combination pattern with mainly of HOXA9 signal in nucleus while a few signals located in the cytoplasm (Figure 1c, 8e and Supplementary Figure 13e).

Q: 5) The western blot data indicate variable HOXA9 protein levels. Why are the cSCCs negative on IHC-IHC is more sensitive than western blotting.

A: Thanks for referee's comments. Actually, the so-called "negative" HOXA9 staining is just extremely faint staining of HOXA9, which looks like to be "negative" when the images are zoomed out, but not truly "negative". We think the emergence of this situation is due to the different sensitivities of the two detection methods. For example, we need to use 1:100 dilution of antibody in IHC while using 1: 1000 ~ 1: 2000 in western blotting. The use of more antibodies is to overcome the problem of lower detection sensitivity of IHC.

Q: 6) The absence of staining in 1C does not correlate with the presence of staining in 1D.

A: Thanks for referee's comment. We are very sorry that we haven't provided a more representative picture in Fig. 1C. As what we have stated in the last question, there is weak staining of HOXA9 in cSCC tumor sections. Now, the staining results have been updated to avoid confusing about the results.

Q: 7) line 214 significance rather than significant.

A: Thanks for referee's suggestion. The error is corrected to be "Based on the significance criterion (Q value), ...".

Q: 8) Gene ontology analysis typically gives very low p values, much lower than 2×10^{-2} . Why is the HIF1a pathway p value not that strong in fig 3D.

A: To address referee's comment, we consulted with experts in bioinformatic analysis and followed their advices to give the following explanations. For GO/Pathway analysis, the Fisher exact test was utilized. The *P* value of Fisher exact test, is determined by the number of significantly-varied genes in GO/Pathway, the number of significantly-varied genes, the number of genes in GO/Pathway and the total number of genes in the annotation system (Y. Benjamini, Y. Hochberg. 1995. Controlling the false discovery rate: a practical and powerful approach to multiple testing. Journal of the Royal Statistical Society, Series B 57 (1): 289–300). In this case, all the *P* value is calculated by a defined formula based on objective data. The ratio of the number of significantly-varied genes in GO/Pathway relative to the number of genes in the same GO/Pathway largely determines the final *P* value. If the *P* value is lower than 0.05, we could not reject the hypothesis that the pathway is functional in HOXA9

knockdown cells. Although the *P* value of the HIF-1 pathway is not as low as the other more pronounced GO terms, it is still below 0.05. Thus, we need to consider its critical importance in HOXA9-mediated molecular events and perform experiments to verify this hypothesis.

Q: 9) Fig 4-a number of letters are missing

A: Thanks for referee's comment. The error of numbering the figure has been corrected.

Q: 10) line 284-85 better to say hematologic malignancies and carcinomas rather than blood cancers or solid cancers

A: Thanks for referee's suggestion. We have corrected the mentioned words as required.

Q: 11) Figs 6a and b could be supplemental.

A: Thanks for referee's suggestion. The original Figure 6a has been moved to Supplementary Fig. 11. Since original Fig. 6b is a strong evidence for supporting the interaction of HOXA9 and CRIP2 in this article, we would like to keep it in Fig. 6 and re-number it to be Fig. 6a.

Q: 12) Fig 6C and D need a CRIP2 peptide control to determine if the association is specific.

A: Thanks for referee's comments. We have added the CRIP2 peptide control to demonstrate the specificity of the association in Fig. 6b,c.

Q: 13) The downregulation of HOXA9 by siCRIP2 in fig 6G is not that significant. How does motility increase in 6C and D with so little change in HOXA9.

A: Thanks for referee's comments. To the best of our knowledge and our findings in this study, CRIP2 does not directly regulate the expression of HOXA9. In this work, we found CRIP2 is the interacting partner of HOXA9 and regulates the roles of HOXA9 through physical association but not changing the expression of HOXA9.

Q: 14) Fig 8e-the staining for HOXA9 could be non-specific. Please clearly should that binding is specific with a HOXA9 peptide preabsorbtion control.

A: Thanks for referee's comments. The pre-absorbed controls demonstrating the staining specificity were performed and added to Fig. 8e and Supplementary Fig. 13e.

Reviewer #2 (Remarks to the Author):

Q: 1 The rescue experiments for the target of miR-365 should be included, in order to prove that the target is really important for the phenotype proposed. This is important as one microRNA has hundreds/thousands of functionally important targets.

A: Thanks for referee's suggestion. We understand the concern and have done the rescue experiments of HOXA9. The results are consistent with the roles of HOXA9 and phenotype proposed in this manuscript. Please check Supplementary Fig. 7 and 10.

Q: 2 Could the authors use some databases of cSCC expression profiling to check if the members of the interactor network have correlated expressions in the way the mechanistic model predicts? This is also an independent assessment of their data obtained in the set of samples presented here.

A: Thanks for referee's suggestion. We have checked several cancer databases like The Cancer Genome Atlas (TCGA) and ICGC Cancer Genome Project. However, no cSCC information can be found in the above two databases. Thus, we checked NCBI GEO for RNA-Seq datasets of cSCC. We have to say that there are limited RNA-Seq datasets can be found and it is hard to compare the expression data between different databases. We choose the GEO dataset GSE66412 from a published article, "Long Noncoding RNA PICSAR Promotes Growth of Cutaneous Squamous Cell Carcinoma by Regulating ERK1/2 Activity." (J Invest Dermatol. 2016 Aug;136(8):1701-10.), for the analysis of correlated expression of our interactor network. After analysis of the expression RPKM data from this dataset, the expression correlation pattern of HOXA9, HIF-1 α and its downstream glycolytic genes is quite consistent with our study and strongly support our HOXA9's mechanistic model that compared with Normal Human Epidermal Keratinocytes (NHEKs), HOXA9 is significantly downregulated in cSCC tumors while HIF-1 α and its downstream glycolytic genes are mainly upregulated. Please check Supplemental Fig. 3.

Q: 3 Western blot result shown on Figure 1A has a n extra sample (HSC-5), but no results.

Were experiments performed in this cell line also?

A: We are very sorry for this careless mistake, which comes from an extra paste operation from an unrelated file. No experiment related to this study has been performed in this cell line.

Q: 4 Figure 2A shows protein expression levels results only, whereas line 185 of manuscript describes "...losses of HOXA9 both mRNA and protein levels...". Gene expression analysis (qPCR results) need to be included in this Figure.

A: Thanks for referee's comments. We have added the qPCR results to Supplementary Fig. 4.

Reviewer #3 (Remarks to the Author):

Q: This is apparently distinct from the mechanism identified previously in hematopoietic stem cells, whereby HOXA9 co-operates with other binding partners to increase HIF-mediated glycolytic gene activation (Kocabas et al. Cell Biosci 2015). It would be important for the authors to discuss their results more equally alongside this previous publication, rather than the slightly biased handling of the paper (page 17, lines 399-402).

A: Thanks for referee's comment. We accept this criticism and revised this part to be "...HOXA9 has been reported to coordinate with both MEIS1 and PBX1 to promote HIF-1 expression in hematopoietic stem cells (HSCs). It is no surprising that HOXA9 always act as an oncogene in hematologic malignancies and positively regulate pro-carcinogenic HIF-1 expression. However, in cSCC, our findings clearly demonstrated that HOXA9 play tumor-

suppressive roles and negatively regulates the expression of HIF-1 α and its downstream glycolytic genes.”

Q: 1. In the introduction, the authors discuss hypoxia and HIF1 stabilization (page 4, final para), particularly with reference to the aetiology of skin cancer (page 5, final para). This is likely to confuse the non-expert reader, due to the fact that this manuscript is essentially concerned with the study of HIF1 in normoxia - i.e. activated by other mechanisms. Although it is accepted that reference to hypoxia is the easiest way to introduce HIF initially, it would perhaps be more relevant for the content of the manuscript if non-hypoxic means of stabilization the alpha subunit were introduced rather than hypoxic.

A: Thanks for referee’s suggestion. We have modified the mentioned paragraphs to meet the referee’s request by emphasizing the non-hypoxic means of stabilization of HIF-1 α . In Page 4, the final paragraph was revised to be “...HIF-1 α is the primary controlling factor in the HIF-1 pathway, acting as the sensor for a hypoxic microenvironment and mediating metabolic responses. Oxygen deprivation inactivates prolyl hydroxylase 2 (PHD2)-mediated hydroxylation and stabilizes HIF-1 α . Under normoxia condition, the presence of oxygen inactivates von Hippel-Lindau (VHL)-mediated proteasomal degradation and thus constitutively activates and stabilizes HIF-1 α . The accumulation of HIF-1 α significantly reduces the efficiency of OXPHOS and promotes glycolysis to enhance cellular survivability under both hypoxic normoxia conditions...” In page 5 final paragraph, the sentence “Oxygenation of the skin epidermis occurs mainly from atmospheric oxygen rather than the vasculature.” has been removed.

Q: 2. GAPDH has previously been shown to be hypoxia-responsive in a number of cell types and is not suitable as a 'housekeeping gene' in hypoxia studies (Zhong et al. Biochem Biophys Res Commun [1999]). If the authors wish to use this single housekeeping gene, they must show unambiguously that its expression is not altered in their systems.

A: Thanks for referee's comments. To confirm if HOXA9-induced events will influence GAPDH expression, the direct evidence is from our transcriptomic analysis of GAPDH expression variation after HOXA9 depletion. GAPDH is not in the gene list of significantly-varied after HOXA9 depletion. A detailed check of the Reads Per Kilobase per Million mapped reads (RPKM) of GAPDH showed that the RPKM value in siHOXA9 sample is 1544.92, just 28.6% more than the RPKM value of 1201.37 in siNC sample, which is not even exceed the lowest 1.5-fold increase criteria for judging the significant influence. Importantly, we have done sufficient protein quantification before the following experiments. From our results, we can guarantee the expression levels of GAPDH is very stable and does not change in response to the variation of HOXA9. Please check Fig. 2a, 5b and Supplementary Fig. 5b, 6b, 7b. Also, as referee pointed out that GAPDH is hypoxia-responsive, our work is about the molecular events happened under normoxia condition. So, we believe the GAPDH is suitably used as 'housekeeping' in this study.

Q: 3. Data from Seahorse experiments is variable and not convincing. For example, in Figure 4E and G, the authors show that the mitochondria in the control cells are already highly uncoupled (compare untreated and FCCP) and the majority of oxygen consumption is not

through the respiratory chain (Rotenone/Antimycin A). This is not consistent with the data in Supplementary Figure 2B, in which a more expected result is shown for the same condition (comparing siNT from each experiment). This suggests that their results are not consistent between experiments, which is concerning and brings into doubt the results they show upon intervention.

A: Thanks for referee's comments. We have consulted the experts and optimized the experimental procedure to keep consistence among different batches of assays. Then we repeated the Seahorse experiments and got consistent results. Now the mentioned seahorse data has been updated which keep the consistency.

Q: 4. In Figure 2F, if apoptosis is hypothesized then this needs to be confirmed with caspase inhibitor, such as zVAD.

A: Thanks for referee's suggestion. We have repeated apoptosis analysis with suggested caspase inhibitor, zVAD, and confirmed our observation definitely to be apoptosis. The updated data has been added to Fig. 2I and Supplementary Fig. 6g.

Q: 5. The knockdown shown in Figure 5A is not more than 50% - it is surprising that this reduction in mRNA can create such a decrease in protein, shown on the right. Is there an effect on mRNA stability as well?

A: Thanks for referee's comments. RNA interference (RNAi) using small interfering RNA (siRNA) inhibits translation or degrade cytoplasmic mRNA by mechanism of post-

transcriptional gene silencing (PTGS). Based on our experience, RNAi using siRNA really have larger impact on protein expression than mRNA level. For referee's question if there is an effect on mRNA stability? To our knowledge, there are several mechanisms is related with mRNA stability. Common decay pathway of most mRNAs is initiated by deadenylation of the 3' poly(A) tail, followed by decapping of the 5'7mGpppN cap and exonucleolytic degradation of the mRNA body in a 5'-3' direction. RNA features with specific sequence and/or structural elements located within 5', 3' UTRs or rare arginine codon in open reading frames (ORFs) have also been suggested to contribute for the decay of a subset of mRNAs. A new report proposed that codon optimality is a major determinant of mRNA stability (Cell. 2015, 160(6):1111-24.). Seeing the complexity of mRNA stability, we have checked the GO/KEGG analysis results and there is only one GO term (GO:0006397~mRNA processing) possibly related with mRNA stability. As the mRNA stability is not likely to be related with the main scope of this study, we will pursue this topic in our future studies.

Q: 6. Figure 5C - the PDK1 blot is not convincing as the band is hardly visible

A: Thanks for referee's comments. We have repeated this experiment and updated the Fig. 5d with clear and convincing data.

Q: 7. The immunoprecipitation experiment, shown in Figure 6D appears not to have worked in the case of the CRIP2 pull-down – as there is little signal for either CRIP2 or HOXA9.

A: Thanks for referee's comments. Although the intensity of signal looks a little weak, significant amount of CRIP2 or HOXA9 can still be detected. Also, the signal intensity is

influenced by the exposure time. To avoid unnecessary misleading of the results, we have repeated the experiment and the picture was updated by a new one with relative stronger signal intensity in Fig. 6c.

Q: 8. The inflection in tumor growth after around 9 days in the absence of HOXA9 (Figure 8A) is very interesting – could the authors discuss this further in the discussion?

A: Thanks for referee's advice. We have discussed this interesting phenomenon in the third paragraph of the Discussion section subtitle "HOXA9 functions as a tumor suppressor in cSCC" as "Among the routine *in vivo* and *in vitro* evaluations of HOXA9 functions, when using xenograft tumor model, there is an inflection in tumor growth after around 9 days in the absence of HOXA9. Checking our previous studies, we found that manipulating the gene expression in xenograft tumors by transfection generally takes about 9 days to observe the visible volume change. In contrast, *in vitro* cellular experiments require only 2-3 days to observe significant changes. Our results indicate the differential sensitivities between *in vivo* and *in vitro* models in response to the variations of specific gene expression."

Q: 9. The knockdown of HOXA9 is also unconvincing in Figure 8E – indeed, the image shown appears to have increased HOXA9. The staining in general in this figure is confusing – HK2 is cytosolic (appears to be nuclear), GLUT1 should be plasma membrane, and PDK1 is mitochondrial – although this wouldn't be visible, it shouldn't be strongly nuclear.

A: To address reviewer's comments, we have checked all the obtained results, repeated some of the IHC detection and updated the mentioned pictures with confirmed results. For

the HOXA9 staining, we can guarantee the HOXA9 expression is really downregulated after rechecking the original sections. To avoid unnecessary misleading, we updated the previous pictures with better ones to show clear difference between knockdown and control samples. For the subcellular localization of HK2, HK2 is generally localized in cytosol in normal cells. However, in cancer cells, HK2 can enter nucleus to perform transcription regulatory role and thus show a mixed pattern by staying in both cytoplasm and nucleus. For GLUT1, our staining results show a really plasma membrane-localized pattern. After check the images in Figure 8e, we understand the figures are too small to show clear pattern of GLUT1. So, we enlarge the figures to make the plasma membrane-localized pattern to be visible. For PDK1, we have checked all the results and confirmed that the subcellular localization of PDK1 in this study is the combination of both cytoplasm and nucleus.

Q: Minor points: 1. Figure 1H blots are not convincing – on the right, both loading control and HOXA9 increase after AntagomiR expression

A: Thanks for referee's comments. We have re-done the experiments and updated the mentioned images in Fig. 1h.

Q: 2. Is the same total cell number represented in each of the scatterplots shown in Figure 2F, as if not, it is difficult to compare?

A: Thanks for referee's comments. The total cell number used for flow cytometry analysis of apoptosis is pre-set to be 10,000 and counted by the machine. We checked the original file and can guarantee the total cell number is equal for all the samples.

Q: 3. No Figure legend is shown in Figure 2H

A: Thanks for referee's comment. There is figure legend of Figure 2H, like "Measurements of cell proliferation by CCK-8 assay (h)".

Q: 4. Contrary to what the authors wrote on page 4, para 2, cancer cells in the main do not 'adopt glycolysis instead of oxidative phosphorylation'. Both are used as without OXPHOS, the appropriate level of anabolism would not be sustained, despite glycolysis. As an example, see papers by Birsoy et al. and Sullivan et al. Cell (2015).

A: Thanks for referee's comments. We are very sorry to arouse this debating. Actually, our original thinking was just to emphasize the importance of glycolysis and didn't exclude the use of oxidative phosphorylation in cancer cells. We have revised this to be '...the efficacy of glycolytic reprogramming is so beneficial and successful that rapidly-proliferating cancer cells tend to strongly enhance the glycolysis while restricting oxidative phosphorylation, regardless of oxygen levels.'

Q: 5. Sentences on page 5 that start, 'PDK1 can inactivate...' (line 91) and 'PDK1 also controls...' line 94 appear to be saying the same thing, and could be simplified.

A: Thanks for referee's suggestion. The two sentences have been revised to be "PDK1 controls the switch of glucose metabolism from aerobic oxidation to glycolysis by inactivating the pyruvate dehydrogenase E1 α subunit, converting pyruvate to acetyl-CoA to inhibit pyruvate metabolism via the tricarboxylic acid cycle¹⁹. Such regulatory role of PDK1 is advantageous to tumor growth because it avoids the accumulation of reactive oxygen species by reducing mitochondrial oxygen consumption^{20, 21}."

Q: 6. It is a little too over-generalized to say that 'Over the past decade, cancer researchers have started to view cancer as a metabolic disorder'

A: Thanks for referee's suggestion. This sentence has been revised to be "Over the past decade, with the emergence of a large amount of evidence supporting the critical roles of glycolytic metabolism in cancers, researchers have started to view cancer as a metabolic disorder⁴¹."

Reviewers' comments:

Reviewer #1 (Remarks to the Author):

The authors have addressed my concerns adequately.

However, there are some minor issues with labeling of figures in supplemental figure 5 and 7. The y axis on the migration and invasion graphs should read "Cell Count" as on other figures.

Reviewer #2 (Remarks to the Author):

Dear Authors,

I appreciate the time to answer the questions asked and perform the experiments that were necessary to perform that task, given their importance for the authenticity of this manuscript. The rescue experiments for HOXA9 were convincing in showing that it is a direct target of miR-365 and it correlates with the phenotype proposed. As stated before, this experiment is of extreme importance, as miRNAs have many targets. I understand the lack of online databases for cSCC, and appreciate the time the authors took to analyze and compare a different publication with the mechanistic model predicted. Nevertheless, it was very helpful in assessing the data obtained in this manuscript. Other questions were responded.

As it was raised by the other reviewers, on line 56, page 3, point number 2: ..."providing rich substrates...". I understand what you mean to say, but it still looks confusing to me. How about changing to: "2) the supply of rich substrates...". Hope it helps. Still, one more time proofreading English should be done, just in case.

Best regards,
George A. Calin.

Reviewer #3 (Remarks to the Author):

The manuscript from Zhou et al. is now much improved. The authors have addressed many of the comments appropriately. However, there are a few remaining concerns:

1. In the revised manuscript, the authors have written more about the control of HIF1 activity in response to a previous request to outline control of HIF1 in normoxia, given the manuscript does not deal with hypoxia. However, what they have written is not always accurate, and a little difficult to understand. Firstly, it is not usual for the alpha subunit of HIF1 to be known as the primary controlling factor in the HIF1 pathway. Its presence is required for a functional heterodimer and its stability is controlled through the PHD enzymes. The authors further state that, 'Under normoxia condition, the presence of oxygen inactivates von Hippel-Lindau (VHL)-mediated proteasomal degradation and thus constitutively activates and stabilizes HIF-1 α .' This is unfortunately factually inaccurate. In normoxia, the PHDs are active, they hydroxylate HIF1 α , which makes it a substrate for the pVHL E3 ubiquitin ligase, resulting in polyubiquitylation and proteasomal degradation. Oxygen itself has not been shown to mediate any effect on pVHL directly, and in normoxia proteasomal degradation is the major means of degrading the alpha subunit – it is active, not inactive. Only under specific conditions is HIF1 α stabilised in normoxia – and if the authors are discussing HIF1 in normoxia, this is what should be introduced. For normoxic stabilisation of HIF1 α , other than conditions where specific genetic lesions are present (e.g. loss of pVHL expression), the authors should consider growth factor signalling, ROS, iron depletion, etc.
2. The addition of zVAD-FMK to confirm whether mode of cell death is caspase-dependent is now there. However, it is not clear where the numbers used in the graph in Figure 2I are coming from – it is not mentioned in the materials and methods or figure legend which quadrant or quadrants are

being used to produce the numbers. Apoptosis can be defined here as either just the top right (late apoptotic cells) or both bottom and top right panels (early and late). This must be clarified.

3. In response to a previous comment (labelled Q.7 in authors' rebuttal), the authors have produced a new panel in Figure 6C, showing a stronger band for CRIP2. However, they have now clipped the western blot significantly to obscure some of the band. Please could a larger section of the gel be shown so that the whole band is visible.

Response to Reviewers' comments:

We greatly appreciate your willingness to consider our revised manuscript, and we especially thank the reviewers for their constructive comments. We have made every effort to address their concerns by carefully proofreading the manuscript to eliminate errors and ambiguities, re-writing several parts of the manuscript, and providing some relevant unpublished data. We also have our manuscript for language polishing by a professional scientist.

We provide below the point-by-point responses to the comments of reviewers, which includes explanations of how we have revised the manuscript to clarify ambiguity and address concerns. For the ease of the reviewers, we have also included quotes from the altered text when appropriate.

We hope that our changes have now made the manuscript suitable for publication. We are more than happy to provide quick response to address any further question.

Reviewer #1: (Remarks to the Author):

Q: The authors have addressed my concerns adequately. However, there are some minor issues with labeling of figures in supplemental figure 5 and 7. The y axis on the migration and invasion graphs should read "Cell Count" as on other figures.

A: Thanks for reviewer's encouragement. For the mentioned minor issues, we have corrected all the labels of y axis on the migration and invasion graphs to be "Cell Count" for consistency as required.

Reviewer #2 (Remarks to the Author):

Q: Dear Authors,

I appreciate the time to answer the questions asked and perform the experiments that were necessary to perform that task, given their importance for the authenticity of this manuscript.

The rescue experiments for HOXA9 were convincing in showing that it is a direct target of miR-365 and it correlates with the phenotype proposed. As stated before, this experiment is of extreme importance, as miRNAs have many targets. I understand the lack of online databases for cSCC, and appreciate the time the authors took to analyze and compare a different publication with the mechanistic model predicted. Nevertheless, it was very helpful in assessing the data obtained in this manuscript. Other questions were responded.

As it was raised by the other reviewers, on line 56, page 3, point number 2: "...” providing rich substrates...”. I understand what you mean to say, but it still looks confusing to me. How about changing to: “2) the supply of rich substrates...”. Hope it helps. Still, one more time proofreading English should be done, just in case.

A: Thanks for reviewer’s encouragement. As suggested by reviewer, we have corrected the mentioned sentence to be “2) the supply of rich substrates for anabolic metabolism of nucleic acids, lipids, and amino acids.” And now, the entire manuscript has been conducted several times of proofreading by experts to ensure the quality of this manuscript.

Reviewer #3 (Remarks to the Author):

The manuscript from Zhou et al. is now much improved. The authors have addressed many of the comments appropriately. However, there are a few remaining concerns:

Q: 1. In the revised manuscript, the authors have written more about the control of HIF1 activity in response to a previous request to outline control of HIF1 in normoxia, given the manuscript does not deal with hypoxia. However, what they have written is not always accurate, and a little difficult to understand. Firstly, it is not usual for the alpha subunit of HIF1 to be known as the primary controlling factor in the HIF1 pathway. Its presence is required for a functional heterodimer and its stability is controlled through the PHD enzymes. The authors further state that, 'Under normoxia condition, the presence of oxygen inactivates von Hippel-Lindau (VHL)-mediated proteasomal degradation and thus constitutively activates and stabilizes HIF-1 α .' This is unfortunately factually inaccurate. In normoxia, the PHDs are active, they hydroxylate HIF1 α , which makes it a substrate for the pVHL E3 ubiquitin ligase, resulting in polyubiquitylation and proteasomal degradation. Oxygen itself has not been shown to mediate any effect on pVHL directly, and in normoxia proteasomal degradation is the major means of degrading the alpha subunit – it is active, not inactive. Only under specific conditions is HIF1 α stabilised in normoxia – and if the authors are discussing HIF1 in normoxia, this is what should be introduced. For normoxic stabilisation of HIF1 α , other than conditions where specific genetic lesions are present (e.g. loss of pVHL expression), the authors should consider growth factor signalling, ROS, iron depletion, etc.

A: Thanks for reviewer's detailed instructions. We are very sorry for not correctly citing the literatures and being misled by unprofessional reports. After carefully reading authoritative

literatures, we have revised the mentioned contents to meet the requirement from reviewer as “Oxygen-responsive HIF-1 α subunit and constitutively-expressed HIF-1 β subunit constitute the heterodimeric HIF-1 transcription factor which plays critical roles in cellular response to hypoxia. Oxygen-sensitive prolyl hydroxylase (PHD) family utilizing oxygen as a co-substrate and iron (Fe²⁺) as a co-factor have four members, each of which possesses a unique role in regulating HIF- α levels, with PHD2 acting as the primary isoform controlling HIF-1 α levels in normoxia. Hydroxylated HIF-1 α can be ubiquitinated by von Hippel–Lindau (pVHL) E3 ubiquitin ligase and targeted for proteasomal degradation. Hypoxia-mediated oxygen deprivation inactivates PHDs and thus stabilizes HIF-1 α . Although HIF-1 α is known owing to its hypoxia-responsive feature, it is regulated by many other factors under normoxia condition such as loss of tumor suppressors, reactive oxygen species (ROS) or oncogene activation. In renal cell carcinoma, loss of pVHL expression owing to genetic lesions of *VHL* gene (mutations or deletions) results in constitutive HIF-1 stabilization. The role of reactive oxygen species (ROS) in HIF-1 α stabilization is more conclusive under normoxic microenvironment. In tumors, oxygen byproducts such as free radicals stabilize HIF-1 α . PI3K/AKT pathway activated by ROS increases HIF-1 α expression and prevents its degradation through increased heat shock protein expression and nitric oxide synthase activation via phosphorylation. It is suggested that iron depletion by ROS oxidization of the iron (Fe²⁺) co-factor inhibits the activity of PHD enzymes and thus contribute to the stabilize HIF-1 α . Further, the growth factor TGF- β 1 acts as an agonist to stabilize HIF-1 α through SMAD pathway-mediated selective inhibition of PHD2 expression.”

Q: 2. The addition of zVAD-FMK to confirm whether mode of cell death is caspase dependent is now there. However, it is not clear where the numbers used in the graph in Figure 2I are coming from – it is not mentioned in the materials and methods or figure legend which quadrant or quadrants are being used to produce the numbers. Apoptosis can be defined here as either just the top right (late apoptotic cells) or both bottom and top right panels (early and late). This must be clarified.

A: Thanks for reviewer comments. The apoptosis in this manuscript is calculated by the sum of both bottom and top right quadrants of the plots, e.g. including both early and late apoptotic cells. We have added this clarification to the Methods/Apoptosis assay section as “Cell apoptosis was detected and quantified using a Guava easyCyte Flow Cytometry System (Merk Millipore) by the sum of both bottom and top right quadrants of the plots, e.g. including both early and late apoptotic cells.”

3. In response to a previous comment (labelled Q.7 in authors’ rebuttal), the authors have produced a new panel in Figure 6C, showing a stronger band for CRIP2. However, they have now clipped the western blot significantly to obscure some of the band. Please could a larger section of the gel be shown so that the whole band is visible.

A: Thanks for reviewer’s comments. We are very sorry for the misleading caused by our experimental operation. Actually, we did not intend to clip the picture to obscure some of the band. The original full picture was shown in Supplementary Figure 16a for reviewer reference. We just wanted to show the bands of HOXA9 and CRIP2 on the same PVDF blot and thus cut the PVDF membrane into two parts for the following detection of HOXA9 and

CRIP2 respectively. However, because the molecular sizes of HOXA9 (30 kDa) and CRIP2 (22 kDa) are too close, this cutting operation at the size of about 26 kDa led to an exposure showing a little clipping of the top of curved CRIP2 bands as the reviewer pointed out. To better reflect the actual situation, we repeated this Co-IP experiments, detected HOXA9 and CRIP2 in two separate intact blots using the same batch of samples and provided new pictures for replacing the previous debating ones in Figure 6c. We also placed the originally-exposed picture to Supplementary Figure 16b for reviewer reference. We wish reviewers could be satisfied by our repeated Co-IP pictures in Figure 6c. Thanks!

REVIEWERS' COMMENTS:

Reviewer #3 (Remarks to the Author):

I thank the authors for the additional work they have now put into their manuscript - it is much improved, and they have addressed all of my concerns.

REVIEWERS' COMMENTS:

Reviewer #3 (Remarks to the Author):

Q: I thank the authors for the additional work they have now put into their manuscript - it is much improved, and they have addressed all of my concerns.

A: Thanks for reviewer's encouragement. It is our responsibility to address all of the reviewers' concerns to make a perfect study.